# Mitigation of CaCO$_3$ Influence on *Ipomoea batatas* Plants Using *Bacillus megaterium* DSM 2894

**Ahmed A. M. Awad** [1,*] , **Alshaymaa I. Ahmed** [2] , **Alaa H. Abd Elazem** [1] and **Atef A. A. Sweed** [1]

1   Soil and Natural Resources Department, Faculty of Agriculture and Natural Resources, Aswan University, Aswan 81528, Egypt; a.hashem@agr.aswu.edu.eg (A.H.A.E.); atefsweed@agr.aswu.edu.eg (A.A.A.S.)
2   Agricultural Microbiology Department, Faculty of Agriculture, Beni-Suef University, Beni-Suef 62511, Egypt; alshaymaa.ibrahim@agr.bsu.edu.eg
*   Correspondence: ahmed.abdelaziz@agr.aswu.edu.eg; Tel.: +20-010-004-211-24

**Abstract:** The application of PGPB is considered a surrogate approach to reducing the amounts of phosphorus fertilizers applied in addition to its role in improving nutrient availability under stress conditions. The objective of this study was to evaluate five levels of calcium superphosphate (CSP); ultimately, CSP was applied in five levels: CSP$_{20}$, CSP$_{40}$, CSP$_{60}$, CSP$_{80}$, and CSP$_{100}$ were applied at 69, 138, 207, 276, and 345 kg ha$^{-1}$, respectively, and two treatments of *Bacillus megaterium* DSM 2894 (with and without) were applied on sweet potato (Beauregard cv.) plants grown in calcareous soils in the 2019 and 2020 seasons in Egypt. Some macro- and micronutrient (i.e., nitrogen (N), phosphorus (P), calcium (Ca), iron (Fe), manganese (Mn), zinc (Zn), and copper (Cu)) uptake, antiradical power (ARP), and protein and total root yields (TRYs) were determined. The plants inoculated with *B. megaterium* DSM 2894 had increased leaf N, P, and Mn contents in both seasons; in addition, Ca was increased in the second season. Furthermore, all of the root nutrient contents (except N) as well as the ARP and TRY were increased in both seasons as compared with those of the noninoculated plants. On the other hand, the maximum values of the leaf Ca, Fe, and Cu contents and the root Ca, Fe, and Zn contents were recorded with the CSP$_{20}$ treatment in both seasons. CSP$_{60}$ was the superior treatment for N (in the leaves), Mn (in the roots), ARP, protein contents, and TRY in both seasons and for the leaf Zn content in the 2019 season. The application of the CSP$_{100}$ treatment gave the highest values for the leaf and root P contents and the root Cu contents in both seasons as well as for the leaf Mn content in the first season and the root N content in the 2020 growth season. Thus, it was concluded that the application of CSP$_{20}$, CSP$_{60}$, and CSP$_{100}$ treatments with the *B. megaterium* DSM2894 mixture gave the best values compared to the use of CSP or DSM2894 individually to attenuate CaCO$_3$-induced damage.

**Keywords:** calcareous soil; phosphorus fertilizer; sweet potato; *Bacillus megaterium* DSM 2894; leaf and tuber nutrient accumulations

## 1. Introduction

Among essential nutrients, phosphorus (P) is one of the main elements of many organic constituents and is vital for many metabolic and physiological processes, such as root elongation and division, photosynthetic ratio, nitrogen fixation enhancement, flowering, and fruit maturation [1]. In the same vein, it is a fundamental element in the energy-rich compounds of living cells such as adenosine triphosphate (ATP), adenosine diphosphate (ADP), and adenosine monophosphate (AMP), as well as in genetic inheritance molecules, such as deoxyribonucleic acid (DNA) and ribonucleic acid (RNA), which are responsible for protein synthesis in plant tissues; neither plants nor animals can grow without them. Furthermore, P is a component of phospholipids, which play a vital role in cellular membranes. P unavailability is one of the major growth-limiting factors affecting plant growth and development resulting from high soil pH, wherein the phosphate ions convert

directly to unavailable forms and become unavailable for uptake by plants [2]. Among these mechanisms, the use of effective agricultural biosystems that take into account the biochemical diversity of entire agricultural ecosystems and their capacity to mitigate the adverse effects of low soil fertility and abiotic stresses, including a high carbonate content in soils and calcareous states, is important [3]. Despite this, P is the least mobile and least available nutrient in soil for plants, especially in calcareous soils [4].

According to FAO 2016 [5], calcareous soils are generally defined as soils that have a high calcium carbonate content ($CaCO_3$) and are characterized by a high soil pH, a low cation exchange capacity (CEC), high salinity (ECe), and soil water retention due to the presence of impermeable crusting layers of $CaCO_3$ on the surface. These layers prevent water percolation to the subsurface layers, increasing the loss of nutrients [6,7] and slowing the release of nutrients such as phosphorus (P), calcium (Ca), and most micronutrients [8,9]. Calcareous soils also suffer from a shortage of organic matter (OM) that affects the growth of crops. Statistically, calcareous soils are common in arid and semiarid regions, occupying >30% of the earth's surface, where precipitation is scarce [10] and $CaCO_3$ content varies considerably [5]. In Egypt, calcareous soils occupy around 270,833 hectares in the northern region of the Western Desert and the Eastern Desert [11]. Under calcareous soil conditions, P is the least mobile and available among all the soil nutrients for plants due to the reaction of P anions with Ca and magnesium (Mg). Furthermore, high pH values form insoluble phosphate compounds and create an imbalance between all elements [12], as P is greatly affected by all calcareous soil properties.

In recent years, due to the increasing prices of phosphorus fertilizers (PFs) and the weak economic status of most farmers in Egypt, in addition to the increased demand for food production, alternatives have been considered. As Egypt is one of the developing countries that suffers from increasing population growth, hindering sustainable agricultural development and expanding the gap between population growth and the availability of agricultural products [13], there is an increasing awareness of the importance of producing healthy food that is free from chemical pollutants [14]. Moreover, these fertilizers contribute to soil degradation in nonarable lands with saline and calcareous characteristics, where most of the applied phosphorus fertilizers accumulate in the soil, while only one-quarter to one-third is utilized [15]. All these reasons have led researchers to work to reduce the amount of PFs applied and to achieve the maximum possible benefit from the used phosphorus fertilizers. Evidence provided from many field studies has indicated the important role of some plant-growth-promoting bacteria (PGPB) in reducing the amount of PFs applied. Among these, PGPB are phosphate-solubilizing bacteria such as *Bacillus* spp., particularly *Bacillus megaterium,* which commonly exists in soil [16].

*Bacillus megaterium* is a subset of plant-growth-promoting rhizobacteria (PGPR) that is widespread in soils and is a member of the microbiome of several plant hosts [17,18]. *B. megaterium* effectively colonizes soils and lives inside plant tissues to facilitate plant growth regulation (PGR) and protection via various mechanisms, directly or indirectly, including increasing phosphorus solubility and nitrogen availability, modulating plant hormones, and producing antimicrobial compounds along with a wide range of bioactive compounds [19,20]. *B. megaterium* is also a source of a wide range of metabolites and enzymes that are involved in PGR [21]. One of the most important characteristics of *B. megaterium* is its ability to improve root development, thus enhancing micronutrient availability [22] and increasing the ability of plants to secrete an array of organic compounds, which attract soil microbes [23], and to maintain mutualistic interactions with roots, thus increasing the ability of crops to grow well and overcome stressful conditions [24,25]. Furthermore, its ability to form spores increases resistance to biotic and abiotic stress conditions [26]. Despite the ability of most PGPB to improve the productivity of different crops, its use on a large scale is still restricted due to the inability of some types to promote plant growth well under different field conditions, particularly under the stress conditions represented by high $CaCO_3$ [27].

These strains are commonly found in most soils worldwide and effectively colonize soils and link with plant root tissues [16–18]. Furthermore, they increase the ability of plants to produce chitinase, which leads to plant resistance against certain pathogens [28]. One of the most dominant rhizosphere bacteria, the *B. megaterium* DSM 2894 strain, has developed different mechanisms to enhance plant growth, directly or indirectly, by increasing the solubility of phosphorus (P) and other micronutrients [29]. In Egyptian soils, tricalcium phosphate, dicalcium phosphate, hydroxy phosphate, and rock phosphate are considered common insoluble inorganic phosphorus fertilizers [30,31]; however, the P in these is present in an unavailable form ($PO_4^{---}$). Many efforts have been made to solubilize the unavailable form into an available form, such as monobasic ($H_2PO_4^{--}$) or dibasic ($HPO_4^{-}$) ions [32]. The DSM 2894 strain can improve plant growth via many mechanisms, in which *B. megaterium* or its metabolites alter the biotic and antibiotic components of the rhizosphere community to bring about plant growth promotion [33,34].

The success of phosphate solubilization depends on the ability of the *B. megaterium* DSM 2894 strain to lower the pH in the surrounding root as a result of the production of $H^+$ ions of organic acids of low molecular weight, including gluconic, formic, 2-ketogluconic, citric, oxalic, lactic isovaleric, succinic, glycolic, and acetic acid, which chelate cations with phosphate [35,36]. In addition to the direct or indirect effects of indole-3-acetic acid [37], the presence of the extracellular oxidation of glucose to gluconic acid via pyrroloquinoline (PQQ) results in a reduction in soil pH. To confirm these results, similar findings were observed in [38], where the production of enzymes, including phosphatase and phytases, that solubilized phosphorus from different sources led to a decreased soil pH in the surrounding roots.

Sweet potato (*Ipomoea batatas* L. Lam.) was selected because it is a staple food in most developing countries, has a short growth period, and is highly adaptive, obtaining a high yield with various climatic changes and low input requirements [39,40]. It is regarded as a strategic crop due to its importance for exports and is considered a food security crop, as it is a superb source of carbohydrates, vitamins, and minerals. In addition, it has been used to treat diseases related to malnutrition in the developing world, including Egypt [41], as well as being utilized as a raw material for many industries, such as flour and starch. Furthermore, it converts more easily to sugars compared with other crops. To date, to the best of our knowledge, the application of DSM 2894 as a PGPB combined with calcium superphosphate (CSP) in the production of sweet potato crops grown in calcareous soil has not been studied. Therefore, the specific objectives of this research were to determine the potential positive effects of *Bacillus megaterium* DSM 2894, individually or combined with CSP, on improving the productivity of sweet potato grown in calcareous soil. In addition to reducing the amount of phosphate fertilizers applied, it could reduce production costs and decrease the environmental pollution resulting from the use of these fertilizers. While this research was motivated by local and regional problems, it will likely have applicability to comparable semiarid areas with calcareous soils.

## 2. Materials and Methods

### 2.1. Study Location, Plant Material, Weather Conditions, and Soil Sampling

Two field experiments were conducted on sweet potato (*Ipomoea batatas* L. cv. Beauregard) plants in sandy loam and sandy clay loam at an experimental farm in the Fayoum district (29°17′ N; 30°53′ E), Egypt, over two consecutive summer seasons in 2019 and 2020. Transplanting was carried out on 10 April 2019 and 19 April 2020 by utilizing vine cuttings. Vines of cv. Beauregard were collected from the AGROFOOD farm on the Alexandria-Cairo Desert Road in the Nubaria district, Egypt. The weather data (April to August) of the study region are provided in Table 1 as an average of both growing seasons. Soil samples of the 0–30 cm surface layer were taken at random from different locations in the study area before transplanting to determine the soil physical and chemical properties, as described in [42]. The soil properties are shown in Table 2.

**Table 1.** Weather data for the study region in Egypt over the sweet potato growing seasons.

| Month | Day $^\circ$C | Night $^\circ$C | RH (%) | AWS (ms$^{-1}$) | AM-PEC-A (mm d$^{-1}$) | AP (mm d$^{-1}$) |
|---|---|---|---|---|---|---|
| | | | 2019 Season | | | |
| April | 34.46 | 9.34 | 49.19 | 2.92 | 5.32 | 0.05 |
| May | 44.23 | 12.04 | 39.88 | 3.58 | 6.67 | 0.00 |
| June | 42.86 | 17.34 | 35.00 | 3.59 | 7.41 | 0.00 |
| July | 41.38 | 18.58 | 37.75 | 3.28 | 6.75 | 0.00 |
| August | 42.52 | 20.69 | 39.56 | 3.63 | 6.72 | 0.00 |
| | | | 2020 Season | | | |
| April | 36.30 | 7.54 | 39.56 | 3.07 | 5.84 | 0.01 |
| May | 45.16 | 12.71 | 27.00 | 3.22 | 7.07 | 0.00 |
| June | 42.45 | 19.41 | 34.50 | 3.76 | 7.71 | 0.00 |
| July | 44.13 | 20.97 | 36.62 | 3.50 | 7.01 | 0.00 |
| August | 42.75 | 21.19 | 38.06 | 3.07 | 6.84 | 0.00 |

Day $^\circ$C = Average day temperature, Night $^\circ$C = Average night temperature, RH = Average relative humidity, AM-PEC-A = Average of measured pan evaporation class A, and AP = Average precipitation.

**Table 2.** Some physical and chemical soil characters of the experimental sites before transplanting in seasons 2019 and 2020.

| Soil Property | 2019 Season | 2020 Season |
|---|---|---|
| Particles size distribution | | |
| Sand % | 63.6 | 66.8 |
| Silt % | 7.8 | 16.5 |
| Clay % | 28.6 | 16.7 |
| Soil texture class | Sandy loam | Sandy Clay loam |
| pH in soil paste | 7.19 | 7.77 |
| ECe (dSm$^{-1}$) in soil paste extracted | 3.95 | 4.24 |
| Soluble ions mmol L$^{-1}$ | | |
| $CO_3^{--}$ | - | - |
| $HCO_3^-$ | 2.03 | 2.70 |
| $Cl^-$ | 21.1 | 25.6 |
| $SO_4^{--}$ | 20.3 | 18.3 |
| $Na^+$ | 31.6 | 31.3 |
| $K^+$ | 0.65 | 0.88 |
| $Ca^{++}$ | 7.11 | 7.47 |
| $Mg^{++}$ | 4.03 | 6.98 |
| Organic matter (OM) % | 0.90 | 1.03 |
| $CaCO_3$ (%) | 10.8 | 11.3 |
| Total N (mg kg$^{-1}$) | 450 | 515 |
| Available-P, mg kg$^{-1}$ (Extractable with NaHCO$_3$(pH = 8.5) | 3424 | 4013 |
| Available K mg kg$^{-1}$ (Extractable with NH$_4$AOC) | 1816 | 1237 |
| Fe, mg kg$^{-1}$ (Extractable with DPTA) | 6.03 | 4.15 |
| Mn, mg kg$^{-1}$ (Extractable with DPTA) | 18.2 | 10.7 |
| Zn, mg kg$^{-1}$ (Extractable with DPTA) | 0.07 | 0.04 |
| Cu, mg kg$^{-1}$ (Extractable with DPTA) | 0.68 | 0.40 |

## 2.2. Treatments and Agricultural Practices

Each of the two field experiments included five levels of super calcium phosphate (Ca$_3$(PO$_4$)$_2 \approx$ 15.5% P$_2$O$_5$; 69 (CSP$_{20}$), 138 (CSP$_{40}$), 207 (CSP$_{60}$), 276 (CSP$_{80}$), and 345 kg (CSP$_{100}$), equal to 20, 40, 60, 80, and 100% of RCSF $\approx$ 345 kg ha$^{-1}$, respectively) with or without inoculation with *Bacillus megaterium* strain DSM 2894 (DSM$_0$ and DSM$_1$, respectively), as shown in Table 3. *Bacillus megaterium* was added as a soil application (drenching) beside the vine cuttings during the first irrigation process. There were 4 rows in each subplot, with each row including 10 cuttings at 30 cm apart, for a total of 40 plants per subplot and a total of 1080 plants. Therefore, the total number of treatments was 10 treatments

($5 \times 2$) in 3 replicates ($10 \times 3 = 30$ plots). The main experimental plots were arranged based on the 5 levels of CSP, and each plot was subsequently subdivided into 2 treatments of *Bacillus megaterium* DSM 2894 (without and with). The agricultural practices for sweet potato cultivation were applied according to the Egyptian Ministry of Agriculture bulletin (No. 1020/2006); however, ammonium sulphate (N% $\approx$ 20.6) as a nitrogen source was added with 432 kg N ha$^{-1}$ in three equal portions in the second, fourth, and seventh weeks after transplanting (WAT). Potassium fertilizer ($K_2SO_4 \approx$ 48–50% $K_2O$) as a potassium source was added at a rate of 230.4 kg ha$^{-1}$, applied in two equal portions in the seventh and ninth WAT. Meanwhile, all treatments of calcium superphosphate were applied during the preparation of the land for transplanting.

**Table 3.** The tested treatment in the study.

| Treatment | Description |
| --- | --- |
| $CSP_{20} \times DSM_0$ | 69 kg ha$^{-1}$ of calcium superphosphate (20% of RPF) with non-inoculated plants by *Bacillus megaterium* DSM 2894 var. |
| $CSP_{20} \times DSM_1$ | 69 kg ha$^{-1}$ of calcium super phosphate (20% of RPF) with inoculated plants by *Bacillus megaterium* DSM 2894 var. |
| $CSP_{40} \times DSM_0$ | 138 kg ha$^{-1}$ of calcium super phosphate (40% of RPF) with non-inoculated plants by *Bacillus megaterium* DSM 2894 var. |
| $CSP_{40} \times DSM_1$ | 138 kg ha$^{-1}$ of calcium super phosphate (40% of RPF) with inoculated plants by *Bacillus megatherium* DSM 2894 var. |
| $CSP_{60} \times DSM_0$ | 207 kg ha$^{-1}$ of calcium super phosphate (60% of RPF) with non-inoculated plants by *Bacillus megaterium* DSM 2894 var. |
| $CSP_{60} \times DSM_1$ | 207 kg ha$^{-1}$ of calcium super phosphate (60% of RPF) with inoculated plants by *Bacillus megaterium* DSM 2894 var. |
| $CSP_{80} \times DSM_0$ | 276 kg ha$^{-1}$ of calcium super phosphate (80% of RPF) with non-inoculated plants by *Bacillus megaterium* DSM 2894 var. |
| $CSP_{80} \times DSM_1$ | 276 kg ha$^{-1}$ of calcium super phosphate (80% of RPF) with inocutated plants by *Bacillus megaterium* DSM 2894 var. |
| $CSP_{100} \times DSM_0$ | 345 kg ha$^{-1}$ of calcium super phosphate (100% of RPF) with non-inoculated plants by *Bacillus megaterium* DSM 2894 var. |
| $CSP_{100} \times DSM_1$ | 345 kg ha$^{-1}$ of calcium super phosphate (100% of RPF) with inoculated plants by *Bacillus megaterium* DSM 2894 var. |

CSP = Calcium SuperPhosphate, RPF = Recommended phosphorus fertilizer.

### 2.3. Experimental Design

The experiments were ordered according to the split-plot structure in a randomized complete block design (RCBD) arrangement with three replicates. Phosphorus fertilizer levels were arranged in the main plots (A), while the inoculation treatments with *Bacillus megaterium* DSM 2894 were arranged in subplots (B). The experimental plot area was 10.5 m$^2$ ($3.5 \times 3.0$ m) and consisted of four rows. Similar top slips (cuttings) of 20 cm lengths were manually planted 25 cm apart on the third top of the slope ridge. *B. megaterium* solution (20 mL) was added to the cuttings during the transplanting process.

### 2.4. Bacterial Strain and Medium Used

*Bacillus megaterium* DSM 2894 was obtained from the Cairo Microbiological Resources Centre (Cairo MIRCEN), Faculty of Agriculture, Ain Shams University, Cairo, Egypt.

Nutrient agar [43] was used for the maintenance of the *B. megaterium* DSM 2894 strain, and it had the following composition (g L$^{-1}$): 5 of peptone and 20 of agar, with the pH adjusted to 7.0. The nutrient broth medium was the same as the nutrient agar without the agar. Glucose broth medium [44] was used for the fermentation process, and it had the

following composition (g $L^{-1}$): 10 of glucose, 3 of peptone, and 3 of beef extract, with the pH adjusted to 7.0.

### 2.5. Inoculum Preparation and Fermentation Process

For the preparation of the standard inoculum, a loop of a single colony of the cultured strain was cultured in a 500-mL plugged Erlenmeyer flask containing 250 mL of nutrient broth and incubated at 30 °C for 24 h on a rotary shaker at 120 rpm. One millilitre of this culture contained $4.8 \times 10^6$ colony-forming units (CFU) and was used as the standard inoculum for the fermentation process. The fermentation process was carried out in plugged Erlenmeyer flasks (10 L in volume) containing 8.5 L of glucose broth medium inoculated with 2% of *B. megaterium* DSM 2894 standard inoculum. The inoculated flasks were incubated at 30 °C for 48 h on a rotary shaker at 120 rpm.

### 2.6. Recorded Data

#### 2.6.1. Nutrient Content of Leaves

At 90 days after transplanting, a random sample of leaves (of five plants) from each experimental unit was collected, washed with distilled water, weighed, oven dried at 70 °C, and weighed again, and the contents of nitrogen, phosphorus, and potassium were determined according to [45]. The total contents of Fe, Mn, Zn, and Cu were determined using inductively coupled plasma–optical emission spectrometry (ICP-OES, Perkin-Elmer OPTIMA-2100 DV, Norwalk, CT, USA) according to the methods described in [46]. The calculation N $\times$ 5.79 (1/0.172 = 6.25) was used to convert nitrogen content to protein content.

#### 2.6.2. Yield and Its Components

At harvest time (120 days after transplanting), all tuberous roots of plants grown in the rows of each subplot were collected and weighed (kg), and data were calculated as the total yield in tons per hectare. A tuberous root sample (10 roots) was randomly chosen from each experimental unit for chemical analysis.

#### 2.6.3. Nutrient Content of Tuberous Roots

At harvest time (120 days after transplanting), 10 uniformly sized tuberous roots from each replicate were collected, cleaned, cut, dried, and ground to determine total N, P, and Ca contents according to the methods described in [46]. The total contents of Fe, Mn, Zn, and Cu were determined as mentioned above. Antioxidant contents were determined according to [47].

### 2.7. The Statistical Analysis

Microsoft Excel 2016 was used to compute the means $\pm$ 1 standard error and to prepare the figures. An analysis of variance for the two seasons was performed based on the split-plot structure in a randomized completed block design (RCBD) using the Genstat statistical package (version 12.1) (VSN International Ltd., Oxford, UK). Means for all variables were separated using Fisher's least significant difference (LSD) test at $p \leq 0.05$ [48].

## 3. Results

### 3.1. Macro- and Micronutrient Contents of Leaves

The influence of five different levels of calcium superphosphate (CSP), inoculation with the *Bacillus megaterium* DSM 2894 strain (DSM 2894), and their interactions on the nitrogen (N), phosphorus (P), and calcium (Ca) contents of sweet potato plant leaves grown under calcareous conditions (CaCO$_3$ 10.8 vs. 11.3%) during the 2019 and 2020 growth seasons are presented in Table 4. The application of CSP had a significant effect on the P and Ca contents of the sweet potato leaf in both seasons, but only had a significant effect on the N content in the 2020 growth season. The results of the interaction indicated that the application of 207 and 345 kg ha$^{-1}$ of CSP (CSP$_{60}$ vs. CSP$_{100}$) has a superior effect on the N (3.97 vs. 3.98%) and P (2.96 vs. 2.94%) contents of the leaf, respectively, compared with the

other applied levels in both seasons, respectively. On the other hand, plants fertilized with 69 kg ha$^{-1}$ of CSP (CSP$_{20}$) had the highest values of leaf Ca (1.008 vs. 1.006%) content in both seasons, respectively. The increase percentages in the 2019 and 2020 growth seasons were, respectively, 8.47 vs. 8.45% for N, 29.26 vs. 27.27% for P, and 34.67 vs. 26.25% for Ca, compared with the lowest values: 3.66 vs. 3.67% for leaf N content and 2.29 vs. 2.31% for leaf P content in the 2019 and 2020 growth seasons, respectively, which were recorded with the CSP$_{20}$ treatment, and 0.75 vs. 0.80% for leaf Ca content in the 2019 and 2020 growth seasons, respectively, which were produced with the CSP$_{100}$ and CSP$_{60}$ treatments.

**Table 4.** Influence of calcium superphosphate (CSP) levels, *Bacillus megaterium* DSM 2894 var. and their interactions on some leaf nutrients content of CaCO$_3$-stressed sweet potato plants in 2019 and 2020.

| Treatment | N | | | P | | | Ca | | |
|---|---|---|---|---|---|---|---|---|---|
| Ca(H$_2$PO$_4$)$_2$ | \multicolumn Leaves (%) | | | | | | | | |
| | DSM$_0$ | DSM$_1$ | Mean | DSM$_0$ | DSM$_1$ | Mean | DSM$_0$ | DSM$_1$ | Mean |
| 2019 Season | | | | | | | | | |
| CSP$_{20}$ | 3.48 [de] ± 0.15 | 3.85 [bc] ± 0.03 | 3.66 [b] ± 0.09 | 1.83 [e] ± 0.06 | 2.74 [b] ± 0.02 | 2.29 [d] ± 0.04 | 0.99 [b] ± 0.01 | 1.03 [a] ± 0.02 | 1.01 [a] ± 0.02 |
| CSP$_{40}$ | 3.40 [de] ± 0.06 | 4.09 [b] ± 0.05 | 3.75 [ab] ± 0.06 | 2.10 [d] ± 0.04 | 2.75 [b] ± 0.03 | 2.42 [cd] ± 0.04 | 0.89 [c] ± 0.02 | 0.91 [c] ± 0.02 | 0.90 [b] ± 0.02 |
| CSP$_{60}$ | 3.34 [e] ± 0.04 | 4.60 [a] ± 0.22 | 3.97 [a] ± 0.13 | 2.37 [c] ± 0.01 | 2.75 [b] ± 0.06 | 2.56 [bc] ± 0.04 | 0.80 [d] ± 0.02 | 0.79 [d] ± 0.02 | 0.79 [c] ± 0.02 |
| CSP$_{80}$ | 3.50 [de] ± 0.03 | 3.95 [bc] ± 0.03 | 3.73 [b] ± 0.03 | 2.36 [c] ± 0.01 | 2.81 [b] ± 0.20 | 2.58 [b] ± 0.02 | 0.79 [d] ± 0.03 | 0.75 [e] ± 0.01 | 0.77 [cd] ± 0.02 |
| CSP$_{100}$ | 3.67 [cd] ± 0.04 | 3.68 [cd] ± 0.06 | 3.68 [b] ± 0.05 | 2.35 [c] ± 0.01 | 3.58 [a] ± 0.07 | 2.96 [a] ± 0.04 | 0.79 [d] ± 0.04 | 0.71 [f] ± 0.03 | 0.75 [d] ± 0.04 |
| Mean | 3.48 [b] ± 0.04 | 4.04 [a] ± 0.08 | 3.76 ± 0.71 | 2.20 [b] ± 0.02 | 2.93 [a] ± 0.02 | 2.56 ± 0.05 | 0.85 [a] ± 0.06 | 0.84 [a] ± 0.02 | 0.85 ± 0.04 |
| 2020 Season | | | | | | | | | |
| CSP$_{20}$ | 3.37 [g] ± 0.11 | 3.97 [c] ± 0.06 | 3.67 [d] ± 0.09 | 1.91 [i] ± 0.01 | 2.72 [e] ± 0.01 | 2.31 [e] ± 0.01 | 0.99 [b] ± 0.03 | 1.02 [a] ± 0.01 | 1.01 [a] ± 0.02 |
| CSP$_{40}$ | 3.38 [fg] ± 0.07 | 4.23 [b] ± 0.09 | 3.81 [c] ± 0.08 | 2.12 [h] ± 0.01 | 2.73 [d] ± 0.02 | 2.43 [d] ± 0.02 | 0.89 [d] ± 0.02 | 0.91 [c] ± 0.01 | 0.90 [b] ± 0.02 |
| CSP$_{60}$ | 3.51 [f] ± 0.09 | 4.46 [a] ± 0.08 | 3.98 [a] ± 0.09 | 2.33 [f] ± 0.01 | 2.75 [c] ± 0.03 | 2.54 [c] ± 0.02 | 0.79 [g] ± 0.02 | 0.80 [g] ± 0.03 | 0.80 [e] ± 0.03 |
| CSP$_{80}$ | 3.65 [e] ± 0.08 | 4.11 [b] ± 0.08 | 3.88 [b] ± 0.08 | 2.32 [f] ± 0.01 | 3.16 [b] ± 0.01 | 2.74 [b] ± 0.01 | 0.79 [h] ± 0.02 | 0.83 [f] ± 0.02 | 0.81 [d] ± 0.02 |
| CSP$_{100}$ | 3.80 [d] ± 0.06 | 3.79 [d] ± 0.07 | 3.80 [c] ± 0.07 | 2.31 [g] ± 0.01 | 3.57 [a] ± 0.01 | 2.94 [a] ± 0.01 | 0.78 [i] ± 0.01 | 0.86 [e] ± 0.02 | 0.82 [c] ± 0.02 |
| Mean | 3.54 [b] ± 0.08 | 4.11 [a] ± 0.08 | 3.83 ± 0.08 | 2.20 [b] ± 0.01 | 2.99 [a] ± 0.02 | 2.59 ± 0.02 | 0.85 [b] ± 0.02 | 0.88 [a] ± 0.02 | 0.87 ± 0.02 |

Mean values (±SE) with different letters in each column are significant (at $p \leq 0.05$). CSP$_{20}$, CSP$_{40}$, CSP$_{60}$, CSP$_{80}$ and CSP$_{100}$ represent CSP added as a soil application at 69, 138, 207, 276, and 345 kg ha$^{-1}$, respectively. DSM$_0$ = non-inoculated with *Bacillus megaterium* DSM 2894 var., DSM$_1$ = inoculated with *Bacillus megaterium* DSM 2894 var., N = nitrogen content, P = phosphorus content and Ca = calcium content.

The results depicted in Figures 1–4 clearly indicate that enhancements in the micronutrient contents of the sweet potato leaf, such as the iron (Fe), manganese (Mn), zinc (Zn), and copper (Cu) contents, were achieved with individual applications of phosphorus fertilizer. The highest content values of Fe (480.9 vs. 479.30 mg kg$^{-1}$) and Cu (23.10 vs. 24.37a mg kg$^{-1}$) were recorded with the application of the minimum level of CSP (CSP$_{20}$) in the 2019 and 2020 growth seasons, respectively. The same treatment (CSP$_{20}$) gave inconsistent results for both the Mn and Zn contents, whereas superior values (91.95 vs. 100.29 mg kg$^{-1}$) for the leaf Mn content were recorded with the application of 138 kg ha$^{-1}$ of CSP (PF$_{40}$) and PF$_{100}$ treatment, as well as for the leaf Zn (26.07 vs. 29.02 mg kg$^{-1}$) content found in the plants treated with CSP$_{60}$ and CSP$_{20}$ in the 2019 and 2020 growth seasons, respectively. The percentage of the increases compared with the lowest values in the 2019 and 2020 growth seasons, respectively, were 21.71 vs. 18.45 for Fe, 11.24 vs. 15.96 for Mn, 17.80 vs. 34.85 for Zn, and 31.40 vs. 22.03% for Cu.

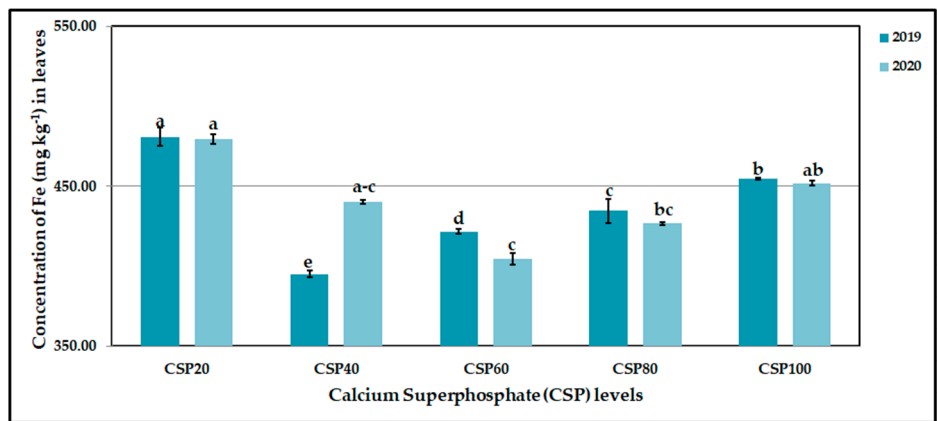

**Figure 1.** Influence of calcium superphosphate (CSP) levels on leaf iron (Fe, mg kg$^{-1}$) contents of CaCO$_3$-stressed sweet potato plants in 2019 and 2020 seasons. Bars with a different letter indicate significant difference between treatments at $p \leq 0.05$. CSP$_{20}$, CSP$_{40}$, CSP$_{60}$, CSP$_{80}$, and CSP$_{100}$ represent CSP added as a soil application at 69, 138, 207, 276, and 345 kg ha$^{-1}$, respectively.

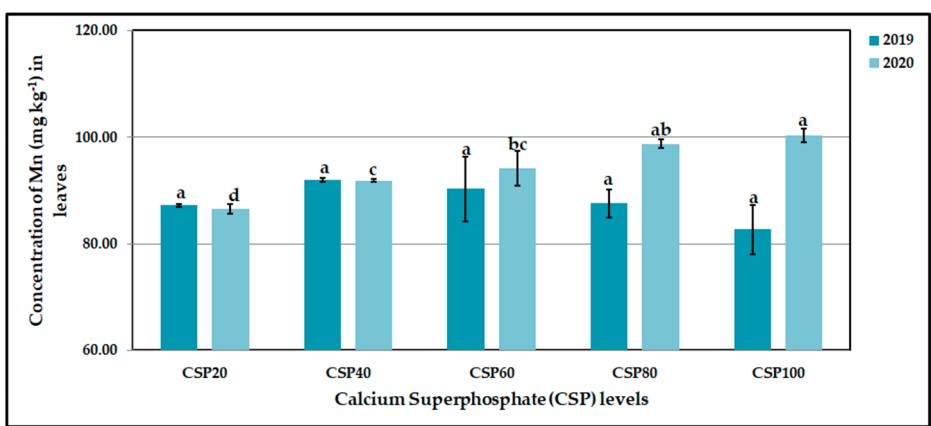

**Figure 2.** Influence of calcium superphosphate (CSP) levels on leaf manganese (Mn, mg kg$^{-1}$) contents of CaCO$_3$-stressed sweet potato plants in 2019 and 2020 seasons. Bars with a different letter indicate significant difference between treatments at $p \leq 0.05$. CSP$_{20}$, CSP$_{40}$, CSP$_{60}$, CSP$_{80}$, and CSP$_{100}$ represent CSP added as a soil application at 69, 138, 207, 276, and 345 kg ha$^{-1}$, respectively.

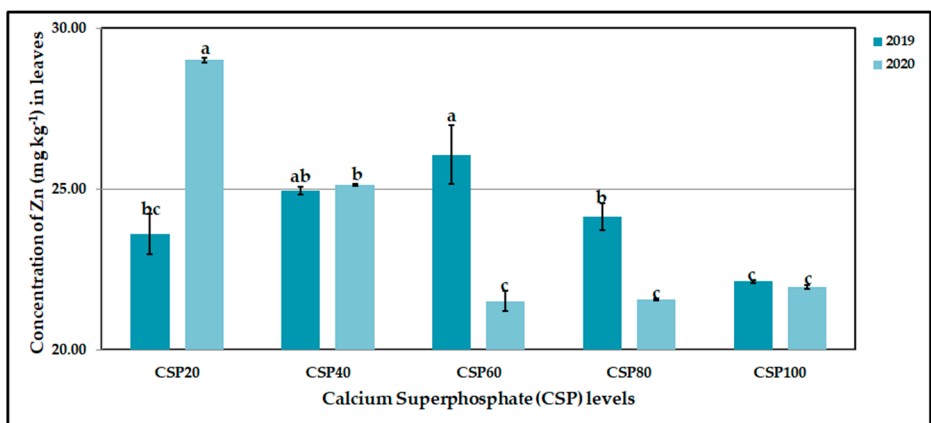

**Figure 3.** Influence of calcium superphosphate (CSP) levels on leaf zinc (Zn, mg kg$^{-1}$) contents of CaCO$_3$-stressed sweet potato plants in 2019 and 2020 seasons. Bars with a different letter indicate significant difference between treatments at $p \leq 0.05$. CSP$_{20}$, CSP$_{40}$, CSP$_{60}$, CSP$_{80}$, and CSP$_{100}$ represent CSP added as a soil application at 69, 138, 207, 276, and 345 kg ha$^{-1}$, respectively.

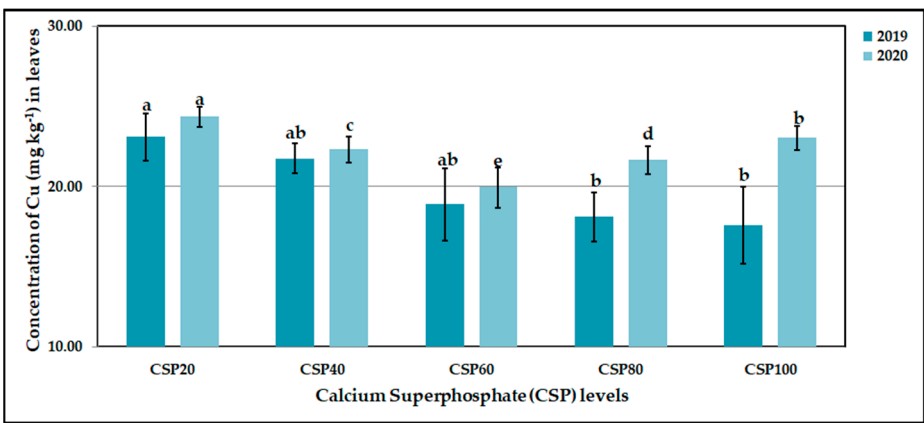

**Figure 4.** Influence of calcium superphosphate (CSP) levels on leaf copper (Cu, mg kg$^{-1}$) contents of CaCO$_3$-stressed sweet potato plants in 2019 and 2020 seasons. Bars with a different letter indicate significant difference between treatments at $p \leq 0.05$. CSP$_{20}$, CSP$_{40}$, CSP$_{60}$, CSP$_{80}$, and CSP$_{100}$ represent CSP added as a soil application at 69, 138, 207, 276, and 345 kg ha$^{-1}$, respectively.

The application of *Bacillus megaterium* DSM 2894 strain (DSM 2894) via individual inoculations resulted in significant increases in the N, P, and Ca contents of sweet potato leaves in the 2019 and 2020 growth seasons, as shown in Table 4. The inoculated plants exhibited significant increases in all the studied macronutrient contents, except for Ca values in the 2019 growth season; however, the maximum values (4.04 vs. 4.11 and 2.93 vs. 2.99%) of the leaf N and P contents were recorded for the plants inoculated with the DSM 2894 strain (DSM$_1$). On the other hand, the highest Ca content (0.85 and 0.88%) was recorded for the non-inoculated sweet potato leaves (DSM$_0$) and the inoculated sweet potato leaves (DSM$_1$) in both growth seasons, respectively. The increase percentages reached 16.09 vs. 16.10 for N and 33.18 vs. 35.91% for P. The decrease percentages were 1.17 vs. 8.59% for the leaf Ca content compared with the lowest values for the three studied nutrients in both growth seasons. There were significant differences in the N, P, and Ca values in both seasons between the inoculated (DSM$_1$) and non-inoculated plants (DSM$_0$).

The influence of the plants inoculated with the DMS 2894 strain on all the above-mentioned micronutrients is shown in Figures 5–8. The results indicated that the micronutrient values in the non-inoculated plants were higher than those in the inoculated plants. The maximum values (491.40 vs. 497.44 for Fe and 24.67 vs. 24.03 mg kg$^{-1}$ for Zn) were produced by the plants not inoculated with the DSM 2894 strain (DSM$_0$) in both seasons; in addition, the leaf Cu content (23.69) reached the maximum value in the 2020 growth season only. On the contrary, the DSM$_1$ treatment gave the maximum values (90.26 vs. 96.88 mg kg$^{-1}$) for the sweet potato leaf Mn content in the 2019 and 2020 growth seasons, respectively, as well as the maximum value for the leaf Cu content (20.00 mg kg$^{-1}$), which was obtained in the inoculated plants in the 2019 growth season. The main effect of DSM 2894 inoculation was statistically significant (at $P \leq 0.01$) for the leaf Fe content in both seasons; however, it was significant for the Zn, Mn, and Cu contents in the 2020 growth season only.

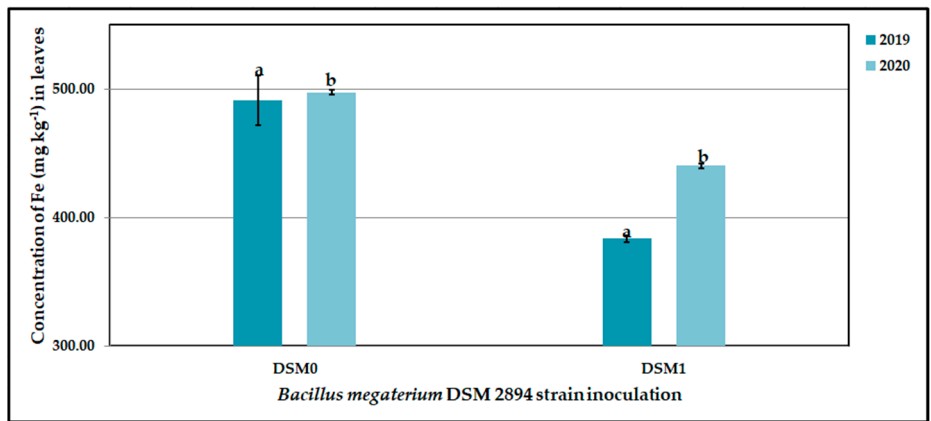

**Figure 5.** Influence of *Bacillus megaterium* DSM 2894 var. on leaf iron (Fe, mg kg$^{-1}$) contents of CaCO$_3$-stressed sweet potato plants in 2019 and 2020 seasons. Bars with a different letter indicate significant difference between treatments at $p \leq 0.05$. DSM$_0$ = non-inoculated with *Bacillus megaterium* DSM 2894 var., DSM$_1$ = inoculated with *Bacillus megaterium* DSM 2894 var.

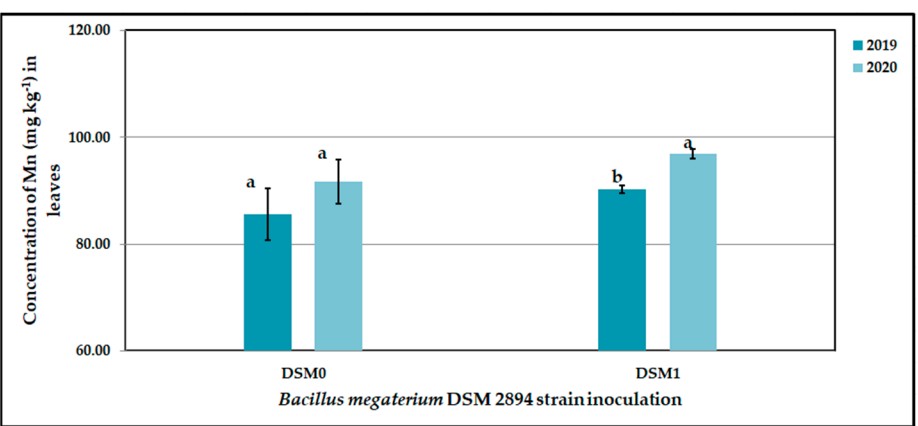

**Figure 6.** Influence of *Bacillus megaterium* DSM 2894 var. on leaf manganese (Mn, mg kg$^{-1}$) contents of CaCO$_3$-stressed sweet potato plants in 2019 and 2020 seasons. Bars with a different letter indicate significant difference between treatments at $p \leq 0.05$. DSM$_0$ = non-inoculated with *Bacillus megaterium* DSM 2894 var., DSM$_1$ = inoculated with *Bacillus megaterium* DSM 2894 var.

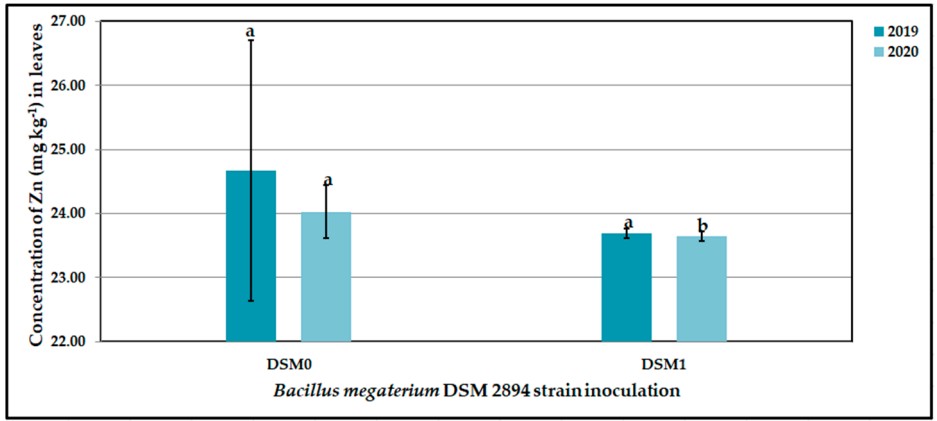

**Figure 7.** Influence of *Bacillus megaterium* DSM 2894 var. on leaf zinc (Zn, mg kg$^{-1}$) contents of CaCO$_3$-stressed sweet potato plants in 2019 and 2020 seasons. Bars with a different letter indicate significant difference between treatments at $p \leq 0.05$. DSM$_0$ = non-inoculated with *Bacillus megaterium* DSM 2894 var., DSM$_1$ = inoculated with *Bacillus megaterium* DSM 2894 var.

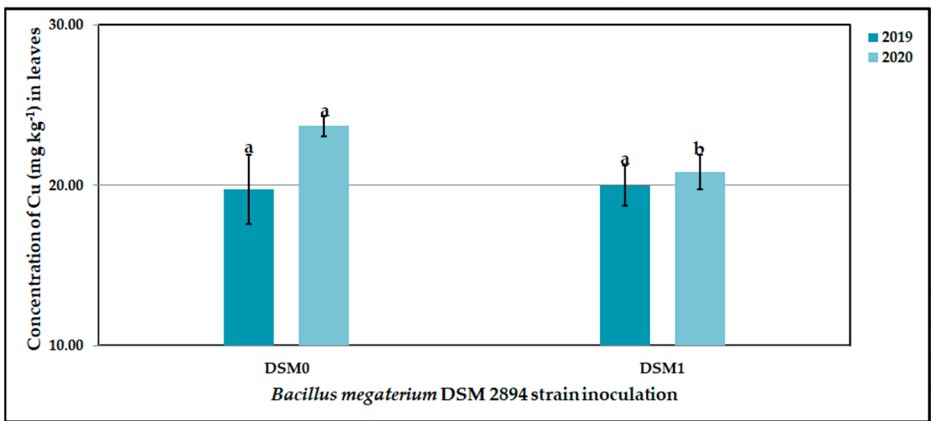

**Figure 8.** Influence of *Bacillus megaterium* DSM 2894 var. on leaf copper (Cu, mg kg$^{-1}$) contents of CaCO$_3$-stressed sweet potato plants in 2019 and 2020 seasons. Bars with a different letter indicate significant difference between treatments at $p \leq 0.05$. DSM$_0$ = non-inoculated with *Bacillus megaterium* DSM 2894 var., DSM$_1$ = inoculated with *Bacillus megaterium* DSM 2894 var.

It is clear from the data given in Table 4 that some of the studied macronutrient contents were significantly affected by the CSP and DSM 2894 interaction in both growth seasons. The obtained results indicate that the application of 207 kg ha$^{-1}$ of CSP with the *B. megaterium* DSM 2894 strain inoculation (CSP$_{60}$ × DSM$_1$) treatment gave the highest values (4.60 vs. 4.46%) of leaf N content, while plants fertilized with 345 kg ha$^{-1}$ of CSP and inoculated with the *B. megaterium* (CSP$_{100}$ × DSM$_1$) treatment produced the maximum (3.58 vs. 3.57%) leaf P values. On the other hand, the plants fertilized with the minimum level of CSP and inoculated with DSM 2894 (CSF$_{20}$ × DSM$_1$) had the highest values (1.03 vs. 1.02%) of leaf Ca content in the 2019 and 2020 growth seasons, respectively.

In this context, the above-mentioned results were corroborated between CSP and DSM 2894 integration for micronutrient contents, as depicted in Table 5. The highest sweet potato leaf micronutrient values reached 531.80 vs. 532.00 for Fe and 30.22 vs. 30.30 mg kg$^{-1}$ for Cu, which were recorded with the application of the CSP$_{20}$ × DSM$_0$ and CSP$_{20}$ × DSM$_1$ treatments for Fe and Cu in the 2019 and 2020 growth seasons, respectively. However, CSP$_{60}$ × DSM$_0$ and CSP$_{100}$ × DSM$_1$ for Mn and CSP$_{60}$ × DSM$_0$ and CSP$_{20}$ × DSM$_0$ for Zn were the superior treatments, recording values of 103.58 vs. 119.51 for Mn and 28.70 vs. 31.48 for Zn in both seasons, respectively. With regard to the lowest values, as shown in Table 5, the minimum values for Fe (336.74 vs. 338.66 mg kg$^{-1}$) and for Cu (13.42 vs. 12.36) were achieved with the application of CSP$_{60}$ × DSM$_0$ for Fe and CSP$_{20}$ × DSM$_0$ for Cu in the 2019 and 2020 growth seasons, respectively. For the lowest leaf Mn (66.70 vs. 73.80 mg kg$^{-1}$) and Zn (20.77 vs. 19.65 mg kg$^{-1}$) values, the data proved that these were achieved with CSP$_{100}$ × DSM$_0$ and CSP$_{20}$ × DSM$_0$ for Mn, and CSP$_{20}$ × DSM$_0$ and CSP$_{60}$ × DSM$_0$ for Zn in both seasons, respectively. The analysis of variance indicated that there were significant variations in the micronutrient contents in both seasons.

**Table 5.** Influence of interaction between phosphorus fertilizer and *Bacillus megaterium* DSM 2894 var. on some leaf micronutrients content of sweet potato plants grown on calcareous saline soil in 2019 and 2020 seasons.

| Treatment | Fe | | Mn | | Zn | | Cu | |
|---|---|---|---|---|---|---|---|---|
| **$Ca(H_2PO_4)_2$** | Leaves (mg kg$^{-1}$) | | | | | | | |
| | DSM$_0$ | DSM$_1$ | DSM$_0$ | DSM$_1$ | DSM$_0$ | DSM$_1$ | DSM$_0$ | DSM$_1$ |
| | 2019 Season | | | | | | | |
| CSP$_{20}$ | 531.80 [a] ± 9.12 | 430.05 [b] ± 2.51 | 74.40 [bc] ± 0.22 | 100.02 [a] ± 0.22 | 20.77 [e] ± 1.24 | 26.46 [ab] ± 0.01 | 15.99 [cd] ± 2.71 | 30.22 [a] ± 0.27 |
| CSP$_{40}$ | 406.90 [b] ± 47.40 | 383.39 [b-d] ± 1.99 | 95.40 [a] ± 0.19 | 88.49 [ab] ± 0.65 | 24.95 [bc] ± 0.21 | 24.95 [bc] ± 0.02 | 21.68 [bc] ± 0.45 | 21.82 [bc] ± 1.42 |
| CSP$_{60}$ | 506.80 [a] ± 24.19 | 336.74 [d] ± 1.47 | 103.58 [a] ± 11.16 | 76.96 [bc] ± 1.09 | 28.70 [a] ± 1.79 | 23.44 [cd] ± 0.02 | 24.35 [ab] ± 1.97 | 13.42 [d] ± 2.57 |
| CSP$_{80}$ | 507.20 [a] ± 13.18 | 362.03 [cd] ± 7.60 | 87.89 [ab] ± 4.84 | 87.23 [ab] ± 0.37 | 25.95 [b] ± 0.80 | 22.35 [de] ± 0.04 | 20.22 [b-d] ± 2.57 | 15.99 [cd] ± 0.44 |
| CSP$_{100}$ | 504.30 [a] ± 1.56 | 405.40 [bc] ± 0.58 | 66.70 [c] ± 7.98 | 98.61 [a] ± 1.17 | 23.00 [c-e] ± 0.02 | 21.26 [de] ± 0.06 | 16.90 [cd] ± 3.14 | 18.56 [b-d] ± 1.69 |
| | 2020 Season | | | | | | | |
| CSP$_{20}$ | 532.00 [a] ± 1.62 | 426.60 [e] ± 4.50 | 73.80 [f] ± 1.06 | 99.18 [c] ± 0.70 | 31.48 [a] ± 0.07 | 26.56 [b] ± 0.07 | 18.44 [e] ± 0.97 | 30.30 [a] ± 0.31 |
| CSP$_{40}$ | 498.20 [b] ± 0.12 | 382.63 [g] ± 2.43 | 94.98 [cd] ± 0.06 | 88.68 [d] ± 0.54 | 25.27 [c] ± 0.03 | 24.97 [c] ± 0.03 | 23.30 [cd] ± 0.48 | 21.33 [d] ± 1.14 |
| CSP$_{60}$ | 470.60 [d] ± 6.35 | 338.66 [i] ± 0.36 | 110.13 [a] ± 6.11 | 78.18 [ef] ± 0.38 | 19.65 [g] ± 0.59 | 23.38 [d] ± 0.01 | 27.58 [b] ± 0.58 | 12.36 [f] ± 1.96 |
| CSP$_{80}$ | 483.60 [c] ± 1.04 | 369.95 [h] ± 1.09 | 98.61 [c] ± 0.63 | 98.85 [c] ± 0.92 | 20.93 [f] ± 0.04 | 22.24 [e] ± 0.02 | 25.76 [bc] ± 0.38 | 17.54 [e] ± 1.34 |
| CSP$_{100}$ | 502.80 [b] ± 0.69 | 401.24 [f] ± 1.82 | 81.06 [d] ± 0.31 | 119.51 [a] ± 2.22 | 22.80 [de] ± 0.09 | 21.10 [f] ± 0.03 | 23.36 [cd] ± 0.76 | 22.72 [d] ± 0.72 |

Mean values (±SE) with different letters in each column are significant (at $p \leq 0.05$). CSP$_{20}$, CSP$_{40}$, CSP$_{60}$, CSP$_{80}$, and CSP$_{100}$ represent CSP added as a soil application at 69, 138, 207, 276, and 345 kg ha$^{-1}$, respectively. DSM$_0$ = non-inoculated with *Bacillus megaterium* DSM 2894 var., DSM$_1$ = inoculated with *Bacillus megaterium* DSM 2894 var., N = nitrogen content, P = phosphorus content and Ca = calcium content.

### 3.2. Tuberous Root Nutrient Contents

The results pertaining to the effect of the phosphorus fertilizer, applied individually, and the plants inoculated with *B. megaterium* are presented in Table 6. The results indicated that the maximum improvements (2.19 vs. 2.44) for tuberous root N contents were found in the plants treated with the CSP$_{100}$ and CSP$_{80}$ treatments in both seasons, respectively. Regarding the uptake of P and Ca, the results revealed that the highest values (4.43 vs. 4.42%) for the P contents were under the CSP$_{100}$ treatment, while for the sweet potato tuberous roots, the Ca contents were recorded (0.35 vs. 1.01%) with three levels of CSP, i.e., CSP$_{20}$, CSP$_{40}$, and CSP$_{60}$ treatments in the 2019 growth season and only the CSP$_{20}$ treatment in the 2020 growth season. The increase percentages reached 11.42 vs. 8.61 for N, 28.89 vs. 28.28% for P, and 11.43 vs. 20.79% for Ca in the 2019 and 2020 growth seasons, respectively. The different levels of phosphorus fertilizer led to significant increases in all the determined nutrients, except for the root N contents in the 2019 growth season.

**Table 6.** Influence of phosphorus fertilizer, *Bacillus megaterium* DSM 2894 var. and their interactions on some tuber nutrients accumulation of CaCO$_3$-stressed sweet potato plants in 2019 and 2020.

| Treatment | N | | | P | | | Ca | | |
|---|---|---|---|---|---|---|---|---|---|
| **$Ca(H_2PO_4)_2$** | Tubers (%) | | | | | | | | |
| | DSM$_0$ | DSM$_1$ | Mean | DSM$_0$ | DSM$_1$ | Mean | DSM$_0$ | DSM$_1$ | Mean |
| | 2019 Season | | | | | | | | |
| CSP$_{20}$ | 2.16 [b] ± 0.01 | 1.73 [d] ± 0.12 | 1.94 [d] ± 0.07 | 2.59 [de] ± 0.00 | 3.70 [d] ± 0.01 | 3.15 [d] ± 0.01 | 0.27 [j] ± 0.03 | 0.43 [a] ± 0.02 | 0.35 [a] ± 0.03 |
| CSP$_{40}$ | 2.19 [b] ± 0.04 | 1.82 [d] ± 0.06 | 2.01 [c] ± 0.05 | 2.72 [g] ± 0.02 | 3.64 [de] ± 0.02 | 3.18 [d] ± 0.02 | 0.27 [j] ± 0.04 | 0.42 [b] ± 0.03 | 0.35 [a] ± 0.04 |
| CSP$_{60}$ | 2.15 [b] ± 0.01 | 1.99 [c] ± 0.04 | 2.07 [b] ± 0.03 | 2.94 [f] ± 0.10 | 3.59 [g] ± 0.03 | 3.27 [c] ± 0.07 | 0.28 [h] ± 0.02 | 0.41 [c] ± 0.06 | 0.35 [a] ± 0.04 |
| CSP$_{80}$ | 2.15 [b] ± 0.06 | 2.15 [b] ± 0.04 | 2.15 [a] ± 0.05 | 3.53 [e] ± 0.10 | 4.21 [b] ± 0.02 | 3.87 [b] ± 0.06 | 0.30 [g] ± 0.03 | 0.36 [d] ± 0.03 | 0.33 [b] ± 0.03 |
| CSP$_{100}$ | 2.07 [bc] ± 0.07 | 2.32 [a] ± 0.04 | 2.19 [a] ± 0.06 | 4.03 [c] ± 0.01 | 4.83 [a] ± 0.01 | 4.43 [a] ± 0.01 | 0.31 [e] ± 0.03 | 0.30 [f] ± 0.03 | 0.31 [c] ± 0.03 |
| Mean | 2.14 [a] ± 0.04 | 2.00 [b] ± 0.06 | 2.07 ± 0.05 | 3.16 [b] ± 0.02 | 3.99 [a] ± 0.02 | 3.58 ± 0.04 | 0.29 [b] ± 0.06 | 0.39 [a] ± 0.02 | 0.85 ± 0.04 |
| | 2020 Season | | | | | | | | |
| CSP$_{20}$ | 2.39 [a] ± 0.11 | 2.13 [bc] ± 0.06 | 2.26 [b] ± 0.09 | 2.57 [j] ± 0.01 | 3.77 [d] ± 0.01 | 3.17 [c] ± 0.01 | 0.99 [b] ± 0.03 | 1.02 [a] ± 0.01 | 1.01 [a] ± 0.02 |
| CSP$_{40}$ | 2.31 [a-c] ± 0.07 | 2.15 [bc] ± 0.09 | 2.23 [b] ± 0.08 | 2.69 [i] ± 0.01 | 3.68 [e] ± 0.02 | 3.18 [c] ± 0.02 | 0.89 [d] ± 0.02 | 0.91 [c] ± 0.01 | 0.90 [b] ± 0.02 |
| CSP$_{60}$ | 2.38 [ab] ± 0.09 | 2.31 [a-c] ± 0.08 | 2.35 [ab] ± 0.09 | 2.80 [h] ± 0.01 | 3.58 [f] ± 0.03 | 3.19 [c] ± 0.02 | 0.79 [g] ± 0.02 | 0.80 [g] ± 0.03 | 0.80 [e] ± 0.03 |
| CSP$_{80}$ | 2.44 [a] ± 0.08 | 2.43 [a] ± 0.08 | 2.44 [a] ± 0.08 | 3.41 [g] ± 0.01 | 4.21 [b] ± 0.01 | 3.81 [b] ± 0.01 | 0.79 [h] ± 0.02 | 0.83 [f] ± 0.02 | 0.81 [d] ± 0.02 |
| CSP$_{100}$ | 2.28 [a-c] ± 0.06 | 2.48 [a] ± 0.07 | 2.38 [ab] ± 0.07 | 4.01 [c] ± 0.01 | 4.84 [a] ± 0.01 | 4.42 [a] ± 0.01 | 0.78 [i] ± 0.01 | 0.86 [e] ± 0.02 | 0.82 [c] ± 0.02 |
| Mean | 2.36 [a] ± 0.08 | 2.30 [a] ± 0.08 | 3.33 ± 0.08 | 3.10 [b] ± 0.01 | 4.01 [a] ± 0.02 | 3.56 ± 0.02 | 0.85 [b] ± 0.02 | 0.88 [a] ± 0.02 | 0.87 ± 0.02 |

Mean values (±SE) with different letters in each column are significant (at $p \leq 0.05$). CSP$_{20}$, CSP$_{40}$, CSP$_{60}$, CSP$_{80}$, and CSP$_{100}$ represent CSP added as a soil application at 69, 138, 207, 276, and 345 kg ha$^{-1}$, respectively. DSM$_0$ = non-inoculated with *Bacillus megaterium* DSM 2894 var., DSM$_1$ = inoculated with *Bacillus megaterium* DSM 2894 var., N = nitrogen content, P = phosphorus content and Ca = calcium content.

The main effect of the phosphorus fertilizer on the tuberous root accumulation of Fe, Mn, Zn, and Cu was significant (at $p \leq 0.05$), as shown in Figures 9–12. The highest accumulation values were 836.50 vs. 853.30 for Fe, 14.60 vs. 14.41 for Mn, 28.37 vs. 28.44 for Zn, and 27.91 vs. 24.67 mg kg$^{-1}$ for Cu, produced with the CSP$_{20}$, CSP$_{60}$, CSP$_{20}$, and CSP$_{100}$ treatments for Fe, Mn, Zn, and Cu in the 2019 and 2020 growth seasons, respectively. The lowest accumulation values were 566.40 vs. 565.20 for Fe, 8.83 vs. 9.06 for Mn, 17.29 vs. 17.08 for Zn, and 16.39 vs. 14.67 for Cu, which were produced with the CSP$_{100}$ treatment for both Fe and Mn, and CSP$_{60}$ for Zn in both seasons. Furthermore, these values were achieved with the CSP$_{20}$ and CSP$_{60}$ treatments for Cu in both growth seasons, respectively. All the previous treatments gave increase percentages in both seasons of 47.69 vs. 50.97 for Fe, 65.35 vs. 59.05 for Mn, 64.08 vs. 66.51 for Zn, and 70.29 vs. 68.17% for Cu as compared with the lowest values in both seasons.

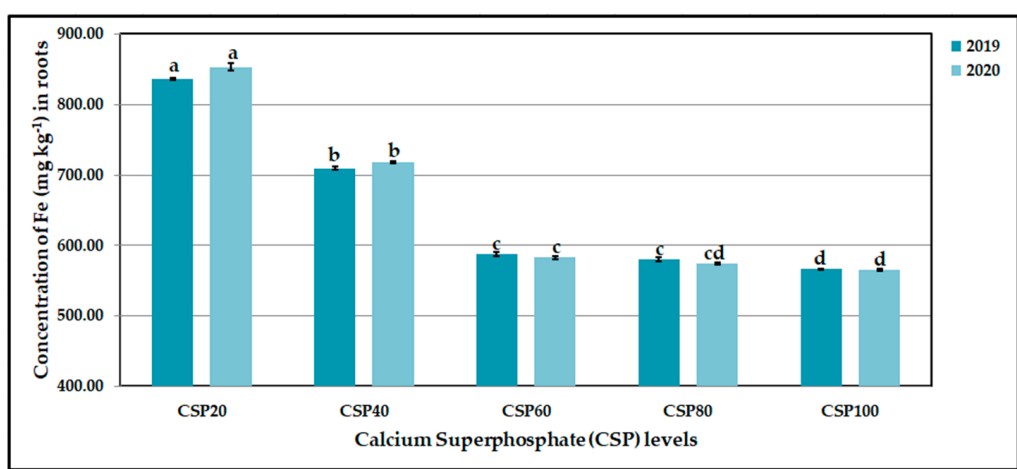

**Figure 9.** Influence of calcium superphosphate (CSP) levels on root iron (Fe, mg kg$^{-1}$) contents of CaCO$_3$-stressed sweet potato plants in 2019 and 2020 seasons. Bars with a different letter indicate significant difference between treatments at $p \leq 0.05$. CSP$_{20}$, CSP$_{40}$, CSP$_{60}$, CSP$_{80}$, and CSP$_{100}$ represent CSP added as a soil application at 69, 138, 207, 276, and 345 kg ha$^{-1}$, respectively.

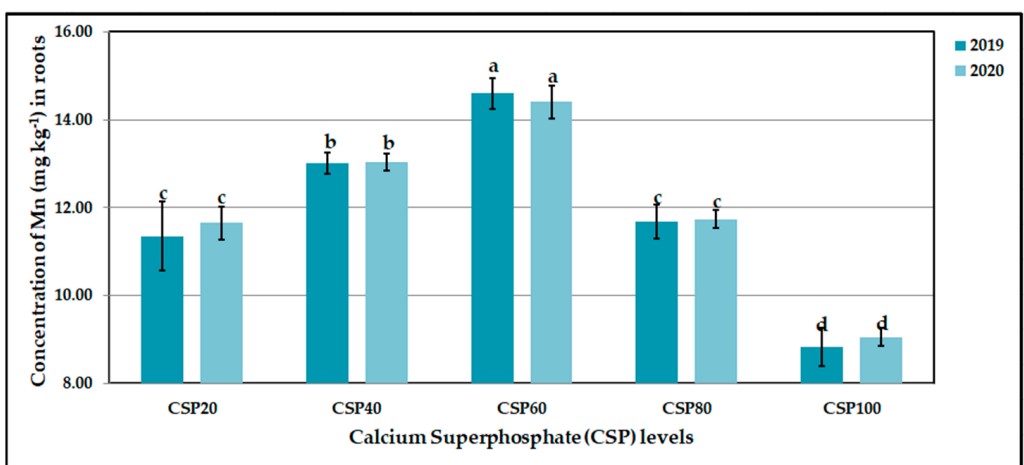

**Figure 10.** Influence of calcium superphosphate (CSP) levels on root manganese (Mn, mg kg$^{-1}$) contents of CaCO$_3$-stressed sweet potato plants in 2019 and 2020 seasons. Bars with a different letter indicate significant difference between treatments at $p \leq 0.05$. CSP$_{20}$, CSP$_{40}$, CSP$_{60}$, CSP$_{80}$, and CSP$_{100}$ represent CSP added as a soil application at 69, 138, 207, 276, and 345 kg ha$^{-1}$, respectively.

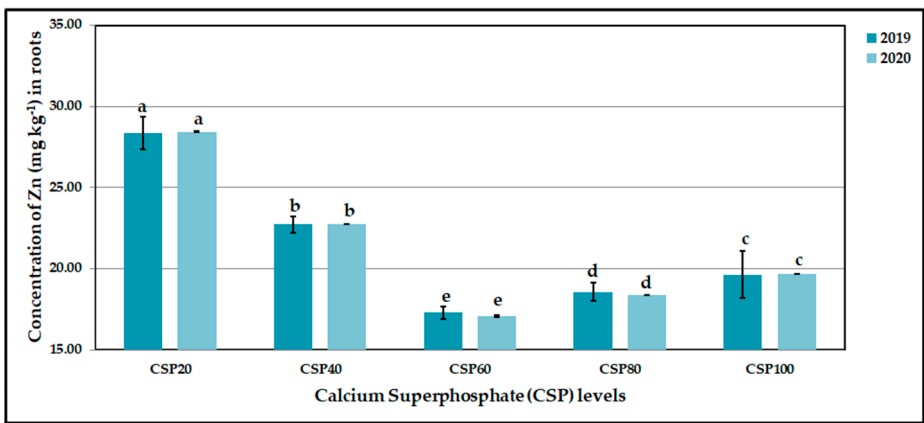

**Figure 11.** Influence of calcium superphosphate (CSP) levels on root zinc (Zn, mg kg$^{-1}$) contents of CaCO$_3$-stressed sweet potato plants in 2019 and 2020 seasons. Bars with a different letter indicate significant difference between treatments at $p \leq 0.05$. CSP$_{20}$, CSP$_{40}$, CSP$_{60}$, CSP$_{80}$, and CSP$_{100}$ represent CSP added as a soil application at 69, 138, 207, 276, and 345 kg ha$^{-1}$, respectively.

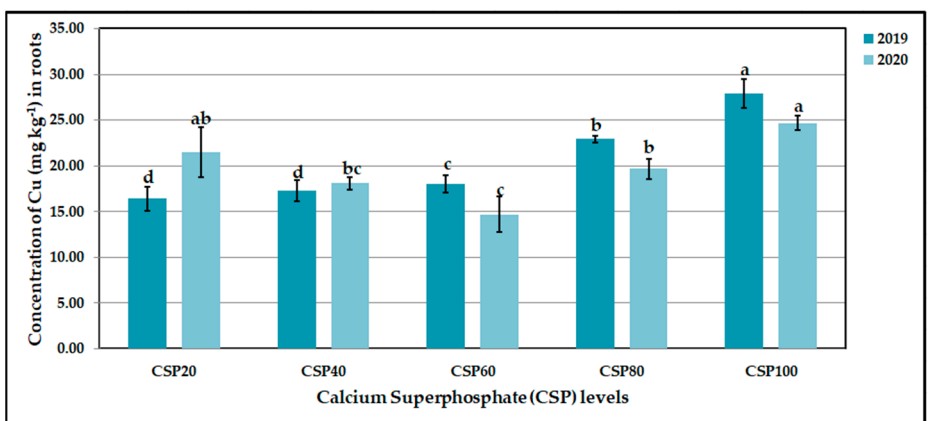

**Figure 12.** Influence of calcium superphosphate (CSP) levels on root copper (Cu, mg kg$^{-1}$) contents of CaCO$_3$-stressed sweet potato plants in 2019 and 2020 seasons. Bars with a different letter indicate significant difference between treatments at $p \leq 0.05$. CSP$_{20}$, CSP$_{40}$, CSP$_{60}$, CSP$_{80}$, and CSP$_{100}$ represent CSP added as a soil application at 69, 138, 207, 276, and 345 kg ha$^{-1}$, respectively.

With regard to DSM 2894 strain influences, the results indicated that the DSM 2894 treatment was the best for sweet potato tuberous root P and Ca accumulation, as mentioned in Table 6. The highest tuberous root (3.99 vs. 4.01%) P and (0.39 vs. 0.88%) Ca accumulation values were recorded in the plants inoculated with the DSM 2894 strain in both seasons. On the other hand, the non-inoculated plants recorded superior values for N accumulation: 2.14 vs. 2.36% in the 2019 and 2020 growth seasons, respectively. The maximum increase percentages reached 6.54 vs. 2.54 for N, 20.80 vs. 22.69 for P, and 25.61 vs. 3.41% for Ca in the 2019 and 2020 growth seasons, respectively. The ANOVA results showed that the tuberous root P and Ca accumulation in both seasons and the N values in the 2019 growth season were significantly affected by inoculation with DSM 2894 bacteria.

The results presented in Figures 13–16 display that the influences of the DSM 2894 strain were excellent in both growth seasons. However, the plants inoculated with the DSM 2894 strain had superior micronutrient accumulation as compared with that of the non-inoculated plants. The maximum values for tuberous roots were 797.26 vs. 797.86 for Fe, 13.55 vs. 13.64 for Mn, 21.83 vs. 21.87 for Zn, and 20.90 vs. 20.82 mg kg$^{-1}$ for Cu in the 2019 and 2020 growth seasons, respectively, which were found in the inoculated plants. Based on the data obtained in our investigation, the increase percentages were 32.29 vs. 33.76 for Fe contents, 24.43 vs. 24.41 for Mn contents, 4.67 vs. 5.58 for Zn contents, and 3.88 vs. 10.57%

for Cu contents in tuberous roots in the 2019 and 2020 growth seasons, respectively. There was a significant influence (at $p \leq 0.01$) of the inoculation with the DSM 2894 strain on tuberous root nutrient accumulation for all the above-mentioned micronutrients.

It is clear from the statistical analysis that the total tuberous root N, P, and Ca contents exhibited significant differences (at $p \leq 0.01$) due to the combination of different levels of CSP and DSM 2894 strain interactions, which influenced sweet potato tuberous root N, P, and Ca accumulation, as presented in Table 6. The $CSP_{100} \times DSM_1$ treatment enhanced the accumulation of N and P, as shown in the following values: 2.32 vs. 2.48 for N and 4.83 vs. 4.84 for P in the 2019 and 2020 growth seasons, respectively. The maximum tuberous root Ca accumulations (0.43 vs. 1.02%) were obtained in the plants with the $CSP_{20} \times DSM_1$ treatment in both seasons. On the contrary, the lowest N and P values in the 2019 and 2020 growth seasons (1.73 vs. 2.13% and 2.59 vs. 2.57, respectively) were produced with the $CSP_{20} \times BM_1$ and $CSP_{20} \times BM_0$ treatments in the 2019 and 2020. For Ca, the $CSP_{20} \times BM_0$ and $CSP_{40} \times BM_0$ treatments in the 2019 growth season and the $CSP_{100} \times BM_0$ treatment in the 2020 growth season gave the minimum values (0.27 vs. 0.78%), respectively. Based on the abovementioned highest and lowest values, the increase percentages were 25.43 vs. 14.11 for N, 46.38 vs. 42.63 for P, and 37.21 vs. 23.53% for Ca in the 2019 and 2020 growth seasons, respectively.

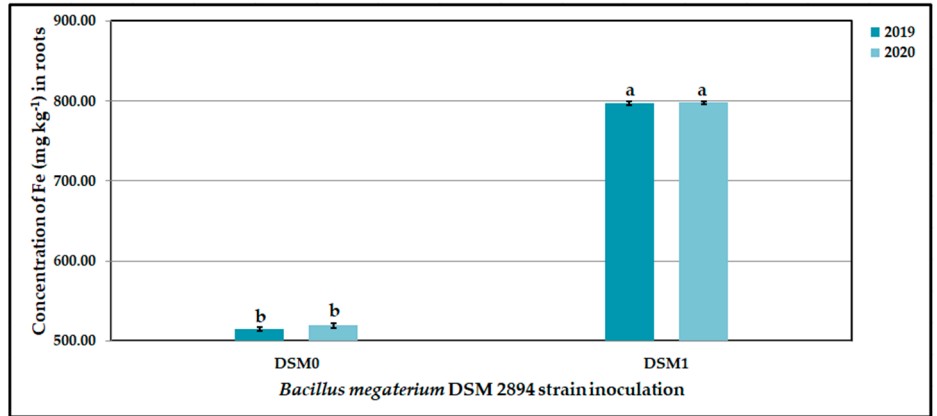

**Figure 13.** Influence of *Bacillus megaterium* DSM 2894 var. on root iron (Fe, mg kg$^{-1}$) contents of CaCO$_3$-stressed sweet potato plants in 2019 and 2020 seasons. Bars with a different letter indicate significant difference between treatments at $p \leq 0.05$. DSM$_0$ = non-inoculated with *Bacillus megaterium* DSM 2894 var., DSM$_1$ = inoculated with *Bacillus megaterium* DSM 2894 var.

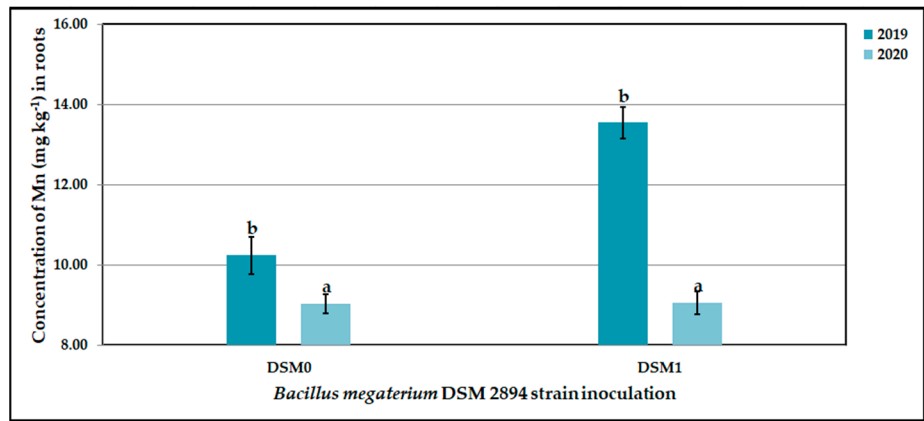

**Figure 14.** Influence of *Bacillus megaterium* DSM 2894 var. on root manganese (Mn, mg kg$^{-1}$) contents of CaCO$_3$-stressed sweet potato plants in 2019 and 2020 seasons. Bars with a different letter indicate significant difference between treatments at $p \leq 0.05$. DSM$_0$ = non-inoculated with *Bacillus megaterium* DSM 2894 var., DSM$_1$ = inoculated with *Bacillus megaterium* DSM 2894 var.

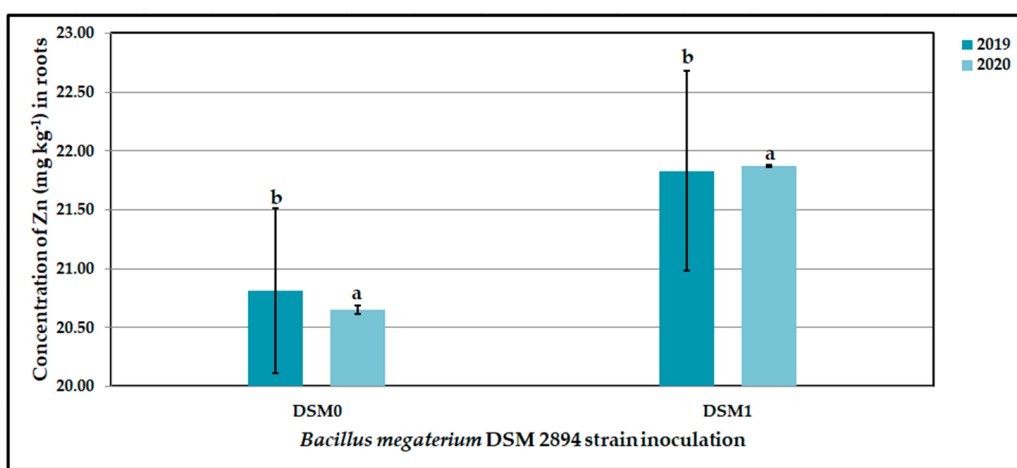

**Figure 15.** Influence of *Bacillus megaterium* DSM 2894 var. on root zinc (Zn, mg kg$^{-1}$) contents of CaCO$_3$-stressed sweet potato plants in 2019 and 2020 seasons. Bars with a different letter indicate significant difference between treatments at $p \leq 0.05$. DSM$_0$ = non-inoculated with *Bacillus megaterium* DSM 2894 var., DSM$_1$ = inoculated with *Bacillus megaterium* DSM 2894 var.

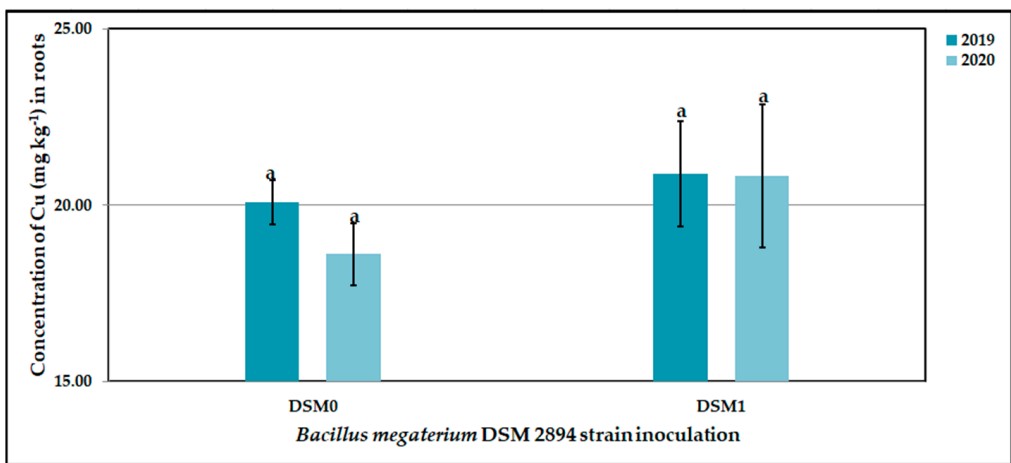

**Figure 16.** Influence of *Bacillus megaterium* DSM 2894 var. on root copper (Cu, mg kg$^{-1}$) contents of CaCO$_3$-stressed sweet potato plants in 2019 and 2020 seasons. Bars with a different letter indicate significant difference between treatments at $p \leq 0.05$. DSM$_0$ = non-inoculated with *Bacillus megaterium* DSM 2894 var., DSM$_1$ = inoculated with *Bacillus megaterium* DSM 2894 var.

Regarding the interaction between the different levels of the CSP treatments and the DSM 2894 strain, the statistical analysis showed significant increase effects on tuberous root Fe, Mn, Zn, and Cu contents, as presented in Table 7. It is evident that the highest accumulation values were 907.80 vs. 946.40 for Fe, 18.94 vs. 18.66 for Mn, 28.94 vs. 29.08 for Zn, and 38.20 vs. 30.30 mg kg$^{-1}$ for Cu and were obtained in plants with the CSP$_{20}$ × DSM$_0$ treatment for Fe and Zn and the CSP$_{60}$ × DSM$_1$ treatment for Mn in the 2019 and 2020 growth seasons, respectively. CSP$_{100}$ × DSM$_1$ and CSP$_{20}$ × DSM$_1$ were the best treatments for Cu in the 2019 and 2020 growth seasons, respectively. The results in this investigation indicated that the inoculation with DSM 2894 did not show any effects on Fe and Zn accumulation.

**Table 7.** Influence of interaction between phosphorus fertilizer and *Bacillus megaterium* DSM 2894 var. on some root micronutrients content of sweet potato plants grown on calcareous saline soil in 2019 and 2020 seasons.

| Treatment | ARP | | Protein | | TRY | |
|---|---|---|---|---|---|---|
| $Ca(H_2PO_4)_2$ | | | (%) | | (Ton ha$^{-1}$) | |
| | | | **Tubers** | | | |
| | **DSM$_0$** | **DSM$_1$** | **DSM$_0$** | **DSM$_1$** | **DSM$_0$** | **DSM$_1$** |
| | | | 2019 Season | | | |
| CSP$_{20}$ | 93.55 [e] ± 4.06 | 183.62 [ab] ± 9.63 | 13.47 [b] ± 1.99 | 19.78 [d] ± 0.74 | 9.03 [f] ± 1.29 | 12.31 [d] ± 1.17 |
| CSP$_{40}$ | 170.16 [bc] ± 6.50 | 157.05 [c] ± 11.41 | 13.70 [b] ± 1.40 | 11.36 [d] ± 1.94 | 10.15 [a] ± 1.07 | 15.09 [ef] ± 0.85 |
| CSP$_{60}$ | 147.84 [c] ± 3.12 | 199.74 [a] ± 16.89 | 13.41 [b] ± 1.64 | 12.41 [c] ± 2.04 | 11.03 [de] ± 1.16 | 18.86 [a] ± 1.97 |
| CSP$_{80}$ | 159.17 [c] ± 4.33 | 169.74 [bc] ± 7.16 | 13.42 [b] ± 0.41 | 13.45 [b] ± 1.89 | 11.86 [d] ± 1.43 | 16.86 [b] ± 2.20 |
| CSP$_{100}$ | 125.83 [d] ± 4.11 | 124.33 [d] ± 6.91 | 12.94 [bc] ± 1.98 | 14.47 [a] ± 1.61 | 12.51 [d] ± 1.47 | 17.24 [b] ± 2.03 |
| | | | 2020 Season | | | |
| CSP$_{20}$ | 95.45 [d] ± 7.44 | 144.96 [bc] ± 11.07 | 14.93 [a] ± 0.86 | 13.32 [a–c] ± 1.51 | 8.90 [d] ± 0.53 | 12.57 [c] ± 2.06 |
| CSP$_{40}$ | 164.15 [a–c] ± 4.20 | 144.95 [bc] ± 6.71 | 14.46 [a–c] ± 1.66 | 13.43 [ab] ± 1.10 | 11.51 [c] ± 1.07 | 15.96 [b] ± 2.41 |
| CSP$_{60}$ | 183.34 [ab] ± 9.62 | 194.45 [a] ± 4.49 | 14.84 [ab] ± 2.32 | 14.41 [a–c] ± 1.07 | 12.24 [a] ± 1.40 | 19.32 [a] ± 1.60 |
| CSP$_{80}$ | 156.25 [a–c] ± 6.01 | 191.37 [a] ± 8.51 | 15.27 [a] ± 2.64 | 15.19 [a] ± 2.02 | 11.38 [c] ± 0.57 | 17.38 [b] ± 2.63 |
| CSP$_{100}$ | 129.17 [cd] ± 8.01 | 188.29 [a] ± 6.76 | 14.27 [a–c] ± 0.71 | 15.49 [a] ± 2.12 | 12.92 [a] ± 1.60 | 16.62 [b] ± 2.04 |

Mean values (±SE) with different letters in each column are significant (at $p \leq 0.05$). CSP$_{20}$, CSP$_{40}$, CSP$_{60}$, CSP$_{80}$, and CSP$_{100}$ represent CSP added as a soil application at 69, 138, 207, 276, and 345 kg ha$^{-1}$, respectively. DSM$_0$ = non-inoculated with *Bacillus megaterium* DSM 2894 var., DSM$_1$ = inoculated with *Bacillus megaterium* DSM 2894 var., N = nitrogen content, P = phosphorus content and Ca = calcium content.

### 3.3. Tuberous Root Yield and Quality

As shown in Table 8 and Figures 17–20, the application of different levels of either CSP or inoculants with the DSM 2894 strain and their interactions affect tuberous root anti-radical power (ARP) and protein contents of sweet potato plants. The maximum values for the tuberous root ARP content in the 2019 and 2020 growth seasons (173.79 vs. 188.89 and 13.70 vs. 15.23, respectively) were obtained in the plants fertilized with CSP$_{60}$ of CSPF individually. For the protein contents, the CSP$_{100}$ and CSP$_{80}$ treatments were the best treatments in the 2019 and 2020 growth seasons, respectively. The different levels of CSP had a significant effect (at $p \leq 0.01$) on the content of ARP in both seasons and the protein contents in the 2019 growth season only. There were no significant differences in protein content in the 2020 growth season.

**Table 8.** Influence of interaction between phosphorus fertilizer and *Bacillus megaterium* DSM 2894 var. on anti-radical power, protein content, and total roots yield of sweet potato plants grown on calcareous saline soil in 2019 and 2020 seasons.

| Treatment | ARP | | Protein | | TRY | |
|---|---|---|---|---|---|---|
| $Ca(H_2PO_4)_2$ | | | (%) | | (Ton ha$^{-1}$) | |
| | | | 2019 Season | | | |
| CSP$_{20}$ | 93.55 [e] ± 4.06 | 183.62 [ab] ± 9.63 | 13.47 [b] ± 1.99 | 19.78 [d] ± 0.74 | 9.03 [f] ± 1.29 | 12.31 [d] ± 1.17 |
| CSP$_{40}$ | 170.16 [bc] ± 6.50 | 157.05 [c] ± 11.41 | 13.70 [b] ± 1.40 | 11.36 [d] ± 1.94 | 10.15 [a] ± 1.07 | 15.09 [ef] ± 0.85 |
| CSP$_{60}$ | 147.84 [c] ± 3.12 | 199.74 [a] ± 16.89 | 13.41 [b] ± 1.64 | 12.41 [c] ± 2.04 | 11.03 [de] ± 1.16 | 18.86 [a] ± 1.97 |
| CSP$_{80}$ | 159.17 [c] ± 4.33 | 169.74 [bc] ± 7.16 | 13.42 [b] ± 0.41 | 13.45 [b] ± 1.89 | 11.86 [d] ± 1.43 | 16.86 [b] ± 2.20 |
| CSP$_{100}$ | 125.83 [d] ± 4.11 | 124.33 [d] ± 6.91 | 12.94 [bc] ± 1.98 | 14.47 [a] ± 1.61 | 12.51 [d] ± 1.47 | 17.24 [b] ± 2.03 |

**Table 8.** *Cont.*

| Treatment | ARP | | Protein (%) | | TRY (Ton ha$^{-1}$) | |
|---|---|---|---|---|---|---|
| Ca(H$_2$PO$_4$)$_2$ | | | | | | |
| 2020 Season | | | | | | |
| CSP$_{20}$ | 95.45 [d] ± 7.44 | 144.96 [bc] ± 11.07 | 14.93 [a] ± 0.86 | 13.32 [a–c] ± 1.51 | 8.90 [d] ± 0.53 | 12.57 [c] ± 2.06 |
| CSP$_{40}$ | 164.15 [a–c] ± 4.20 | 144.95 [bc] ± 6.71 | 14.46 [a–c] ± 1.66 | 13.43 [ab] ± 1.10 | 11.51 [c] ± 1.07 | 15.96 [b] ± 2.41 |
| CSP$_{60}$ | 183.34 [ab] ± 9.62 | 194.45 [a] ± 4.49 | 14.84 [ab] ± 2.32 | 14.41 [a–c] ± 1.07 | 12.24 [a] ± 1.40 | 19.32 [a] ± 1.60 |
| CSP$_{80}$ | 156.25 [a–c] ± 6.01 | 191.37 [a] ± 8.51 | 15.27 [a] ± 2.64 | 15.19 [a] ± 2.02 | 11.38 [c] ± 0.57 | 17.38 [b] ± 2.63 |
| CSP$_{100}$ | 129.17 [cd] ± 8.01 | 188.29 [a] ± 6.76 | 14.27 [a–c] ± 0.71 | 15.49 [a] ± 2.12 | 12.92 [a] ± 1.60 | 16.62 [b] ± 2.04 |

Mean values (±SE) with different letters in each column are significant ($p \leq 0.05$). CSP$_{20}$, CSP$_{40}$, CSP$_{60}$, CSP$_{80}$, and CSP$_{100}$ represent CSP added as a soil application at 69, 138, 207, 276, and 345 kg ha$^{-1}$, respectively. DSM$_0$ = non-inoculated with *Bacillus megaterium* DSM 2894 var., DSM$_1$ = inoculated with *Bacillus megaterium* DSM 2894 var., ARP = Anti-radical power and TRY = Total roots yield.

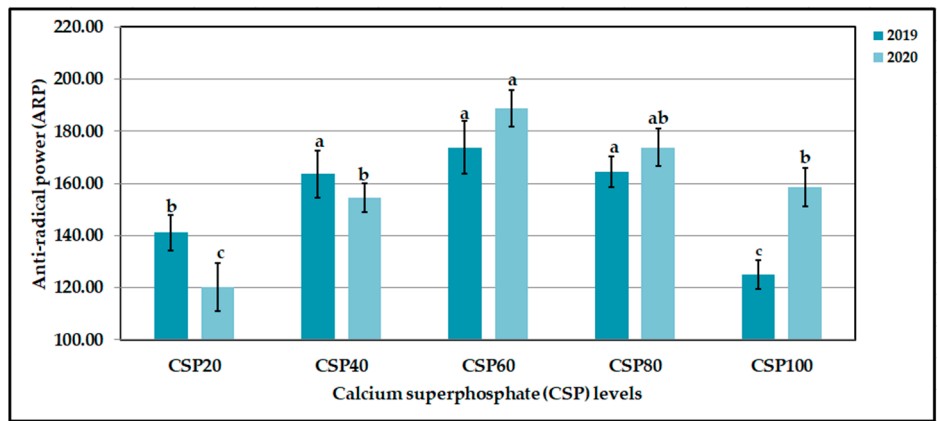

**Figure 17.** Influence of calcium superphosphate (CSP) levels on anti-radical power (ARP) of tubers of CaCO$_3$-stressed sweet potato plants grown in 2019 and 2020 seasons. Bars with a different letter indicate significant different between treatments at $p \leq 0.05$. CSP$_{20}$, CSP$_{40}$, CSP$_{60}$, CSP$_{80}$, and CSP$_{100}$ represent CSP added as a soil application at 69, 138, 207, 276, and 345 kg ha$^{-1}$, respectively.

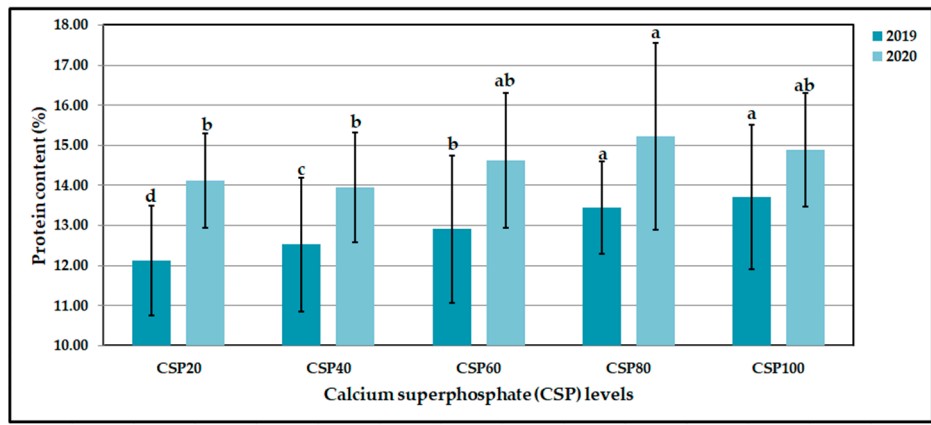

**Figure 18.** Influence of calcium superphosphate levels (CSP) on protein content of CaCO$_3$-stressed sweet potato plants grown in 2019 and 2020 seasons. Bars with a different letter indicate significant different between treatments at $p \leq 0.05$. CSP$_{20}$, CSP$_{40}$, CSP$_{60}$, CSP$_{80}$, and CSP$_{100}$ represent CSP added as a soil application at 69, 138, 207, 276, and 345 kg ha$^{-1}$, respectively.

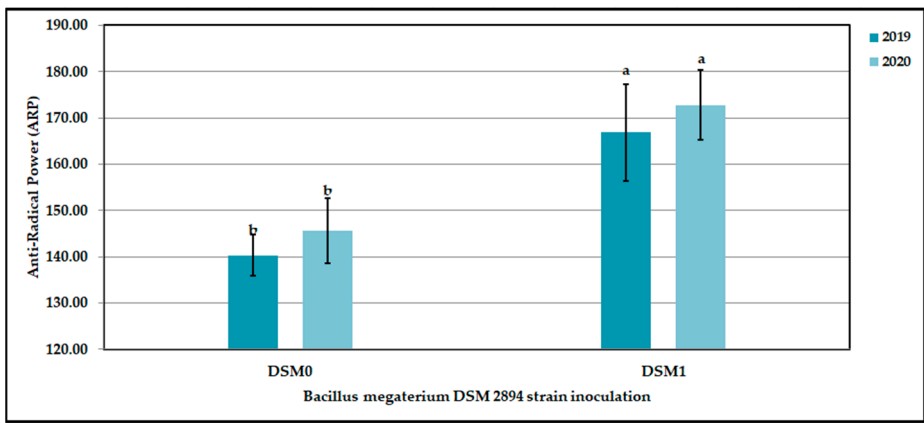

**Figure 19.** Influence of *Bacillus megaterium* DSM 2894 var. on anti-radical power (ARP) of tubers of $CaCO_3$-stressed sweet potato plants grown in 2019 and 2020 seasons. Bars with a different letter indicate significant different between treatments at $p \leq 0.05$. $DSM_0$ = non-inoculated with *Bacillus megaterium* DSM 2894 var., $DSM_1$ = inoculated with *Bacillus megaterium* DSM 2894 var.

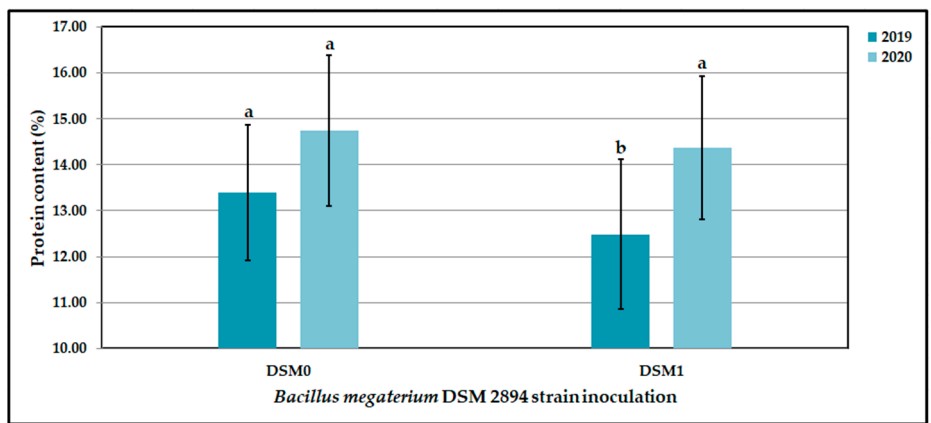

**Figure 20.** Influence of *Bacillus megaterium* DSM 2894 var. on protein content (%) of tubers of $CaCO_3$-stressed sweet potato plants grown in 2019 and 2020 seasons. Bars with a different letter indicate significant different between treatments at $p \leq 0.05$. $DSM_0$ = non-inoculated with *Bacillus megaterium* DSM 2894 var., $DSM_1$ = inoculated with *Bacillus megaterium* DSM 2894 var.

With respect to the influence of the DSM 2894 inoculation, it is clear from Figures 18 and 20 that the superior values were 166.89 vs. 172.80 for ARP and 13.39 vs. 14.75% for protein, which were produced in the inoculated ($DSM_1$) plants for ARP and non-inoculated ($DSM_0$) plants for protein contents in 2019 and 2020 growth seasons, respectively. Inoculation treatment had a significant effect (at $p \leq 0.01$) on ARP contents in both seasons and protein contents in the 2019 growth season, while there were no significant differences for protein content in the 2020 growth season.

The influence of the mixture of phosphorus fertilizer with *B. megaterium*, as shown in Table 8, showed the superiority of this treatment ($CSP_{60} \times DSM_1$) regarding ARP values in both seasons. However, the highest values (14.47 vs. 15.49%) were found with the treatment of $CSP_{100} \times DSM_1$ and $CSP_{80} \times DSM_1$ in both seasons, respectively. Statistical analysis of variance indicated that combined treatment led to a highly significant increase in antioxidant and protein contents in the 2019 growth season, while it was non-significant in the 2020 growth season.

All the treatments had a significant effect on the tuberous root yield of the sweet potato as a result of the different levels of CSP. The results depicted in Figure 21 indicate the maximum values (14.95 vs. 15.78 followed by 14.88 vs. 14.77 tons ha$^{-1}$) of the tuberous root yield of sweet potato, which were obtained using $CSP_{60}$ and $CSP_{100}$ treatments in the

2019 and 2020 growth seasons, respectively. Furthermore, the increase percentages in the 2019 and 2020 growth seasons were, respectively, 40.11 and 47.06% as compared with the lowest values (10.67 vs. 10.73 ton ha$^{-1}$), which were obtained with the CSP$_{20}$ treatment. Regarding the influence of DSM 2894 on the tuberous root yield, as shown in Figure 22, the inoculated plants had the highest values (16.07 vs. 16.37 tons ha$^{-1}$) of the tuberous roots in the 2019 and 2020 growth seasons, respectively. In the comparison to the non-inoculated plants, the increase percentages of the inoculated plants reached 47.16 vs. 43.72 in the 2019 and 2020 growth seasons, respectively.

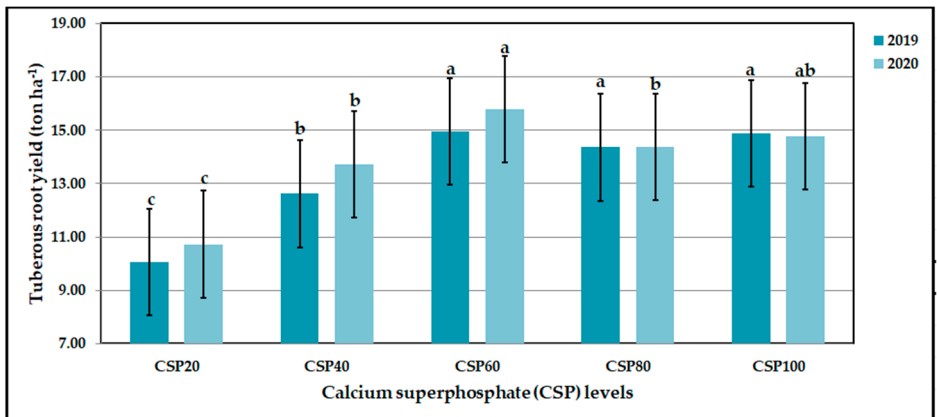

**Figure 21.** Influence of calcium superphosphate (CSP) levels on total roots yield of CaCO$_3$-stressed sweet potato plants grown in 2019 and 2020 seasons. Bars with a different letter indicate significant different between treatments at $p \leq 0.05$. DSM$_0$ = non-inoculated with *Bacillus megaterium* DSM 2894 var., DSM$_1$ = inoculated with *Bacillus megaterium* DSM 2894 var.

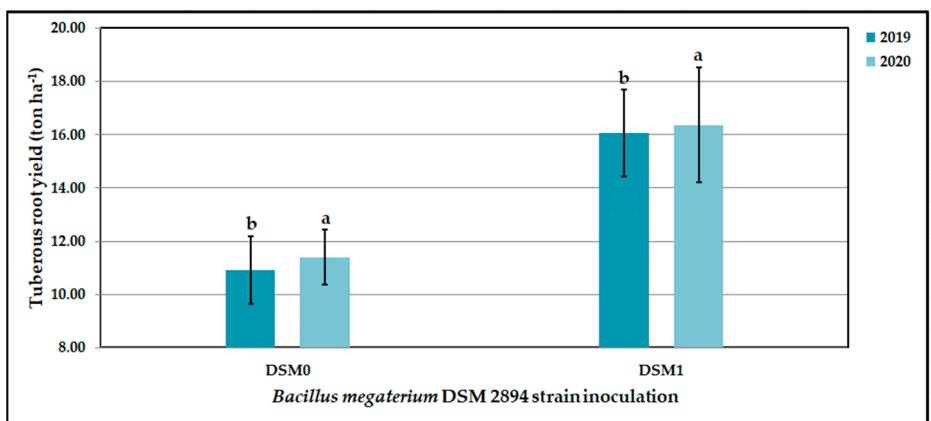

**Figure 22.** Influence of *Bacillus megaterium* DSM 2894 var. on total root yield of CaCO$_3$-stressed sweet potato plants grown in 2019 and 2020 seasons. Bars with a different letter indicate significant different between treatments at $p \leq 0.05$. DSM$_0$ = non-inoculated with *Bacillus megaterium* DSM 2894 var., DSM$_1$ = inoculated with *Bacillus megaterium* DSM 2894 var.

The results of the different levels of CSP with the DSM 2894 strain interactions are presented in Table 8. The maximum tuberous root yield (18.86 vs. 19.32 ton ha$^{-1}$ in the 2019 and 2020 growth seasons, respectively) was recorded for the CSP$_{60}$ × DSM$_1$ treatment. On the other hand, the lowest tuber root yield (9.03 vs. 8.90 ton ha$^{-1}$) in the 2019 and 2020 growth seasons, respectively, was recorded for the CSP$_{20}$ × DSM$_0$ treatment. The results of the statistical analysis showed that the CSP treatment with the DSM 2894 interactions had a significant (at $p \leq 0.01$) influence on the total tuberous root yield of sweet potato in both seasons.

## 4. Discussion

The soil tested in this study had undesirable characteristics, such as a low organic matter content (SOM 0.90 vs. 1.03%), a high pH (7.19 vs. 7.77), a high calcium carbonate content ($CaCO_3$ 10.8 vs. 11.3%), and a high salinity content (ECe 3.95 vs. 4.24 dSm$^{-1}$), as shown in Table 1. This resulted in low fertility with a nutritional imbalance that affected the nutrient uptake and performance of the grown sweet potato plants. Many strains of plant growth–promoting bacteria (PGPB) such as *Bacillus* spp., including *Bacillus megaterium*, have been applied to solubilize insoluble phosphate and to decrease the quantity of phosphorus fertilizers used.

As shown in Tables 4 and 5, the enhanced nitrogen and phosphorus contents of the sweet potato leaves may be due to the fundamental role of the DSM 2894 strain in enhancing the amino acid content, suggesting an increase in the bioavailability of N and P in the soil and their accumulation in the plant [37]. In other words, *Bacillus* spp. play a vital role in enhancing macro and micronutrients in the soils, thereby improving their uptake by plants [25,49]. The improved influence of the *Bacillus* strain on the uptake of nutrients may be attributed to *Bacillus* markedly increasing the protein synthesis via activation of the enzyme nitrate reductase [50]. These results were confirmed by [51] using a PGPR consortia involving *B. megaterium* that enhanced the protein contents of chili and cauliflower. The results pertaining to N and P uptake may be explained by the DSM 2894 strain in soil decreasing the soil pH in the surrounding roots due to the production of organic acid as a secondary metabolite that can overcome some of the undesirable soil characteristics [52], as shown in Table 2. These results are in accordance with the findings of [53] regarding common bean plants. On the contrary, some leaf nutrient contents such as calcium (Ca), as presented in Table 4, were decreased by higher levels of phosphorus fertilizer, which may be due to higher levels inhibiting the production of gluconic acid by *B. megaterium* [36]. In other words, low levels of P, as presented in CSP$_{20}$ with inoculation with Bacillus, interact Ca with phosphate ions in the soil to form insoluble calcium phosphate, which, along with CSP fertilizer, contains calcium within its chemical structure $Ca(H_2PO_4)_2$; however, in the case of small amounts, the Ca can be linked with phosphate ions in the soil to form insoluble calcium phosphate.

The direct effect of *B. megaterium* on some leaf micronutrients (i.e., iron (Fe), manganese (Mn), zinc (Zn), and copper (Cu) contents) may involve biological processes as well as the solubility of complex organic and inorganic nutrients; the mobilization of Fe via the production of siderophores; and plant growth regulator production, such as indole acetic acid (IAA), gibberellin, and cytokinins [33]. This results in improvements in plant growth and nutrient uptake through either direct or indirect mechanisms and their metabolites altering the biotic and abiotic components of the rhizosphere community to bring about plant growth promotion [30]. These results are consistent with those of other studies, such as those in [54], in terms of leaf Mn and Cu uptake. These results have been described by some researchers as many microorganisms having the ability to produce phytohormones, including varieties of auxin (i.e., indole-3-acetic acid (IAA), indole -3-butryric acid (IBA), indole-3-pyruvic acid (IPA), indole lactic acid (ILA), and tryptophol (TOL)), cytokinins, and gibberellin acid, which affect some physiological and morphological processes, regulating plant growth via increasing the root volume and modifying the root system, resulting in a higher nutrient uptake from the soil. In their trials, the authors of [55] suggested that a *Bacillus* strain could solubilize tricalcium phosphate fertilizers by lowering the soil pH value due to its production of organic acids. These findings can be supported by our results; however, decreasing the soil pH affected the micronutrients' solubility (Table 5). In addition, improvements in the leaf nitrogen (N) and phosphorus (P) contents as a result of the *Bacillus* DSM 2894 strain affected N$_2$-fixation bacteria (Table 4); N and P are essential nutrients for the growth and reproduction of crops. Accordingly, our results related to the yield indicated that the total root yield (TRY) in the 2019 and 2020 growth seasons rose from 14.95 and 15.78 to 18.86 and 19.32 ton ha$^{-1}$, respectively, and that the percentage increase in the 2019 and 2020 growth seasons reached 26.15 and 22.43%, respectively, compared with the individual

application of phosphorus fertilizer (CSP$_{60}$ treatment), as shown in Table 8. In another words, *Bacillus* DSM 2894 improved the soil's chemical, physical, and biological properties and increased the microbial activities of rhizobacteria [56]. *B. megaterium* enhanced plant growth via several mechanisms, such as releasing volatile substances; [57–59] pointed out that 2-pentylfuran produced by *B. megaterium* XTBG 34 as a volatile substance enhanced the growth of *Arabidopsis thaliana*. These results were confirmed by [60,61]; 2R and 3R butanediol produced by the *B. sublilis* JS strain improved the biomass of *A. thaliana* by stimulating the phytohormone auxin, which can directly or indirectly promote plant growth, thereby enhancing the nutrient uptake that was reflected in the total root yield.

These results are in line with those of studies [62–64] on crops such as kiwifruits, peas, and strawberries. The results were also confirmed by [65], as the microorganisms caused significant increases in root (increase percentage exceeding 90%) and shoot growth (an increase of approximately 50%) compared with non-inoculated plants, leading to an increase in nutrient uptake. The antagonism between Fe $\times$ Mn and Fe $\times$ Zn is shown in Table 5, and may also be related to ferric-chelate reductase activity [64]. These results are in agreement with other findings described by [26,66]. On the other hand, the synergistic interactions (Fe $\times$ P, P $\times$ Zn, Cu $\times$ Fe, and Cu $\times$ P) may be due to the influence of macronutrients on root reductase activity [67].

## 5. Conclusions

The findings of our study showed that the integrative application of *Bacillus megaterium* DSM 2894 and calcium superphosphate fertilizer (CSP) was more effective than the individual application of each in alleviating defective soil characteristics and in enhancing the responses of sweet potato plants to calcareous (CaCO$_3$ = 10.8 and 11.3%) stress damage. Sweet potato plants grown in the integrative DSM 2894 + CSP-treated calcareous soil maintained higher contents of N, P, Ca, and Cu (in the leaves), and N, P, Ca, Mn, and Cu (in the roots), as well as higher antiradical power (ARP), protein, and total root yield. It can be concluded from these results that this integrative treatment played a pivotal role in repairing defective soil properties, as positively reflected in the sweet potato plant growth, development, and metabolism, and in the responses to calcareous stress. It can also be concluded that DSM 2894 + CSP can be used as a form of integrated plant nutrition under normal or abnormal conditions.

**Author Contributions:** Conceptualization, A.A.M.A. and A.A.A.S.; data curation, A.A.M.A. and A.A.A.S.; formal analysis, A.A.M.A. and A.I.A.; investigation, A.A.M.A., A.A.A.S. and A.I.A.; methodology, A.A.M.A. and A.I.A.; resources, A.A.M.A. and A.I.A.; software, A.I.A. and A.A.A.S.; writing—original draft, A.A.M.A.; writing—review and editing, A.A.M.A.; A.H.A.E. and A.A.A.S. and A.I.A. All authors have read and agreed to the published version of the manuscript.

**Funding:** This research received no external funding.

**Institutional Review Board Statement:** Not applicable.

**Informed Consent Statement:** Not applicable.

**Data Availability Statement:** The data presented in this study are available upon request from the corresponding author.

**Conflicts of Interest:** The authors declare no conflict of interest.

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
