# Peer review of "Mitigation of CaCO3 Influence on Ipomoea batatas Plants Using Bacillus megaterium DSM 2894"

_agronomy, doi:10.3390/agronomy12071571_

Round 1
Reviewer 1 Report
In this study authors investigated "Mitigation of CaCO3 influence on Ipomoea batatas plants using Bacillus megaterium DSM 2894 var.". The work is important, but the manuscript suffers from lack of clarity, the methods are inadequately described, the data analysis appears inadequate, and the results presented do not provide adequate support for the conclusions.
Author Response
Agronomy - MDPI
Manuscript ID: agronomy-1756752
Manuscript Title: " Mitigation of CaCO3 Influence on Ipomoea batatas Plants Using Bacillus megaterium DSM 2894 var."
===================================================================================
Dear Ms. Evelyn Zhang
Section Managing Editor   Â
Thank you for your efforts and I’d like also to thank very much the reviewers for their valuable comments. We have corrected the manuscript based on the comments of reviewers, and the corrections made in the text in red color, and are outlined step by step as follows:
Response to the comments of Reviewer 1:
Open review
(x ) I would not like to sign my review report
(Â Â )Â I would like to sign my review report
English language style
(x ) Extensive editing of English language and style required
( Â Â ) Moderate English changes required
(Â Â ) English language and style are fine/minor spell check required
(Â Â ) I don't feel qualified to judge about the English language and style
|
Yes |
Can be improved |
Must be improved |
Not applicable |
|
|
Does the introduction provide sufficient background and include all relevant references? |
( Â Â ) |
(x) |
( Â Â ) |
(Â Â ) |
|
Are all the cited references relevant to the research? |
(Â Â ) |
(x ) |
( Â Â ) |
( Â Â ) |
|
Is the research design appropriate? |
(Â Â ) |
(x ) |
( Â Â ) |
( Â Â ) |
|
Are the methods adequately described? |
(Â Â ) |
(x ) |
( Â Â ) |
(Â Â ) |
|
Are the results clearly presented? |
( Â Â ) |
(x) |
( Â Â ) |
( Â Â ) |
|
Are the conclusions supported by the results? |
(Â Â ) |
(x) |
( Â Â ) |
( Â Â ) |
Comments and suggestions for authors
In this study authors investigated "Mitigation of CaCO3 influence on Ipomoea batatas plants using Bacillus megaterium DSM 2894 var.". The work is important, but the manuscript suffers from lack of clarity, the methods are inadequately described, the data analysis appears inadequate, and the results presented do not provide adequate support for the conclusions.
Re 1. Regarding English language style. Dear reviewer, I would like to inform you that our manuscript have been undergone for reviewing via MDPI editing service. (Invoice No.42590 dated on April 3, 2022) and based on your suggestion, our manuscript was send again to English language editing (invoice NO 46167 dated on June 19, 2022). Â
Re 2. Part from discussion was added to the introduction section to elucidate the role of Bacillus spp in details as shown with red color (line 104-129)
Re 3. We reviewed all cited references and we made sure of that.
Re 4. This research includes 2 factors, the first factor was application of different levels of CSP (the least important) and put as mean, while the second factor was Bacillus megaterium (the most important) that put as a sub-mean. Therefore, this study ordered according to split-plot design to know the effect of CSP and Bacillus individually and their interaction. All these were written in materials and methods section (line 182-189).
Re 5. The results were rewritten again. I hope is satisfied for you, and I am ready to make any other required modifications from your point of view.
Re 6. Of course , the conclusion was supported with the obtained results.      Â
Many thanks to Reviewer 1 for his valuable comments
Ahmed A. M. Awad (Corresponding author)
June 22, 2022Â

Reviewer 2 Report
The study is interesting and valuable since it addresses an important issue of Mitigation of CaCO3 Influence on Ipomoea batatas Plants Using Bacillus megaterium DSM 2894 Var.–
The research methodology is appropriate, the experiment was properly designed and described in sufficient detail, including the characteristics of soil and climate conditions; agronomic conditions
The results generally are clearly presented (including the statistical significance of research findings), and they provide new information on the potential improvements in the performance of calcareous soils.
An in-depth discussion of the results is based upon carefully selected - but I have some comments in this chapter.
My comments are presented below.
Introduction
I think there is too much information on phosphorus itself. Nearly half of this chapter is about phosphorus. Bacillus megaterium does more than just produce phosphorus.
Line 49. "There is an ..." - this is probably not true.
Line 58 "characterized by high soil acidity (pH), - this is probably not true
Line 63 "... that affects the growth of croplands ..." - I don't get it
Line 79 "... Moreover, there are negative influences of these fertilizers on soil degradation .." - firstly, not all phosphorus fertilizers have a negative effect on the soil, and secondly, I did not find such information in the quoted work [15]. The mentioned authors [15] deal with manure.
line 118. - Whether this statement is the purpose of the work - I do not think. It is rather a wish
Results
The results are interesting, but too many numbers in the text and the reader may get lost.
1) Line 219, I consider it unnecessary to enter, e.g. the results are presented in table 4. the same in other places
2) I consider it superfluous to multiply the subsections: 3.1.2., 3.1.3. 3.2.2, 3.2.3 etc. Main chapters are enough
3) Line 221. Please check the interpretation of the results. There is a terrible confusion here. CSP 60 significantly influenced the highest N content and CSP 100 the highest Ca. In my opinion, this text cannot be read in such a way.
Line 265. In my opinion, in Table 4, it is stated that in 2019 it did not significantly affect N (LSD), and neither did Ca in 2020 (LSD). Why are these values when only homogeneous groups are described?
Table and figures
Please enter under the tables which means *
There is no explanation of what they refer to (factor or factor * year)
Why there is no analysis for years.
I don't understand inserted homogeneous groups in figures (some of them).
e.g. Fig. 1 - marking with the letter "c" is higher than "ab" and "e", it should rather be between them
Some citations are poorly structured and read badly, e.g.
line 198 ... methods described in [34].
Line 192 ... the methods described by [35]
the same line 195, 635, 640
Discussion
Line 554-586
In this area, this is not a discussion, a description that could be included in the Introduction
List of References
You have to format correctly, spaces, unnecessary dots etc.
Author Response
Agronomy - MDPI
Manuscript ID: agronomy-1756752
Manuscript Title: " Mitigation of CaCO3 Influence on Ipomoea batatas Plants Using Bacillus megaterium DSM 2894 var."
===================================================================================
Dear Ms. Evelyn Zhang
Section Managing Editor   Â
Thank you for your efforts and I would like also to thank very much the reviewers for their valuable comments. We have corrected the manuscript based on the comments of reviewers, and the corrections made in the text in red color, and are outlined step by step as follows:
Response to the comments of Reviewer 1:
Open review
(Â ) I would not like to sign my review report
(x)Â I would like to sign my review report
English language style
( ) Extensive editing of English language and style required
( ) Moderate English changes required
( ) English language and style are fine/minor spell check required
(x) I don't feel qualified to judge about the English language and style
|
Yes |
Can be improved |
Must be improved |
Not applicable |
|
|
Does the introduction provide sufficient background and include all relevant references? |
( Â Â ) |
(x) |
( Â Â ) |
(Â Â ) |
|
Are all the cited references relevant to the research? |
(x) |
( Â Â ) |
( Â Â ) |
( Â Â ) |
|
Is the research design appropriate? |
(x) |
( Â Â ) |
( Â Â ) |
( Â Â ) |
|
Are the methods adequately described? |
(x) |
( Â Â ) |
( Â Â ) |
(Â Â ) |
|
Are the results clearly presented? |
( Â ) |
(x) |
( Â Â ) |
( Â Â ) |
|
Are the conclusions supported by the results? |
(x) |
( Â ) |
( Â Â ) |
( Â Â ) |
Based on your comments regarding improve the introduction and results, we would like to inform you that some modifications have been made in both sections as shown with a red colorÂ
|
Comments and suggestions for authors |
||
|
· The study is interesting and valuable since it addresses an important issue of Mitigation of CaCO3 Influence on Ipomoea batatas Plants Using Bacillus megaterium DSM 2894 Var. |
||
|
||
|
· The results generally are clearly presented (including the statistical significance of research findings), and they provide new information on the potential improvements in the performance of calcareous soils |
||
|
||
|
Introduction I think there is too much information on phosphorus itself. Nearly half of this chapter is about phosphorus. Bacillus megaterium does more than just produce phosphorus |
||
|
1 |
Line 49. "There is an ..." - this is probably not true |
this sentence was removed |
|
2 |
Line 58 "characterized by high soil acidity (pH), - this is probably not true |
corrected as shown in line 57 |
|
3 |
Line 63 "... that affects the growth of croplands ..." - I don't get it |
this sentence was corrected as shown in lines 61-63 |
|
4 |
Line 79 "... Moreover, there are negative influences of these fertilizers on soil degradation .." - firstly, not all phosphorus fertilizers have a negative effect on the soil, and secondly, I did not find such information in the quoted work [15]. The mentioned authors [15] deal with manure |
this sentence was carefully reviewed and cited from Whalen and Chang, 2001. In research titled “Effects of the continuous use of organic manure and chemical fertilizer on soil inorganic phosphorus fractions in calcareous soil†by song et al., 2017  |
|
5 |
line 118. - Whether this statement is the purpose of the work - I do not think. It is rather a wish |
Indeed, however I noted “it will probably …… |
|
Results The results are interesting, but too many numbers in the text and the reader may get lost. |
||
|
6 |
Line 219, I consider it unnecessary to enter, e.g. the results are presented in table 4. the same in other places |
Done |
|
7 |
I consider it superfluous to multiply the subsections: 3.1.2., 3.1.3. 3.2.2, 3.2.3 etc. Main chapters are enough |
Done |
|
8 |
Line 221. Please check the interpretation of the results. There is a terrible confusion here. CSP60 significantly influenced the highest N content and CSP100 the highest Ca. In my opinion, this text cannot be read in such a way |
we made a slight modification as shown from line 222-224 and the results were discussed in lines 584-588 and 597-602 |
|
9 |
Line 265. In my opinion, in Table 4, it is stated that in 2019 it did not significantly affect N (LSD), and neither did Ca in 2020 (LSD). Why are these values ​​when only homogeneous groups are described? |
we described the results in full because it is fair to present the effect of the treatments, whether it was a positive or negative effects. My dear reviewer, the negative effect in itself is a result that must be clarified, such as the result of the statistical analysis, whether it is significant or not significant   |
|
Table and figures |
||
|
10 |
Please enter under the tables which means * |
I removed all the LSD values while keeping Duncan’s letter, therefore, there is no need to define * (in all tables) |
|
11 |
There is no explanation of what they refer to (factor or factor * year) |
I specialize in the field of plant nutrition, so the factors of the study are the levels of phosphate fertilizers, and the most important factor is PGPS in addition, their interactions  |
|
12 |
Why there is no analysis for years. |
years are not a study factor, but the conduct of the second season (20200 was to confirm the study |
|
13 |
I don't understand inserted homogeneous groups in figures (some of them). e.g. Fig. 1 - marking with the letter "c" is higher than "ab" and "e", it should rather be between them |
indeed, there are intended mistake and all letters have been reviewed according to statistical analysis |
|
14 |
Some citations are poorly structured and read badly, e.g. line 198 … methods described in [34] |
Done in line 188 |
|
15 |
Line 192 ... the methods described by [35] the same line 195, 635, 640 |
Some citations (36 and 37) were removed. Done in lines 191. Please, lines 635 don’t contain any citation. The citations in line 640 were modified as shown in lines 626 and 627.  |
|
Discussion |
||
|
16 |
Line 554-586. In this area, this is not a discussion, a description that could be included in the introduction |
Done |
|
17 |
List of References You have to format correctly, spaces, unnecessary dots etc. |
Done |
Many thanks to Reviewer 1 for his valuable comments
Ahmed A. M. Awad (Corresponding author)
June 22, 2022

This manuscript is a resubmission of an earlier submission. The following is a list of the peer review reports and author responses from that submission.
Round 1
Reviewer 1 Report
The work "Mitigation of CaCO3 influence on Ipomoea batatas plants using Bacillus megaterium DSM 2894 var." has the potential to be significant, however the text lacks clarity, the methods are poorly defined, the data analysis appears to be inadequate, and the results and discussion presented do not fully support the conclusions. The following are some examples of specific comments that demonstrate these points:
I have found that the authors did not revise this manuscript according to previous comments (as below):
Title
I suggest rewriting the tittle of the manuscript.
Abstract
Line 10-14: “Under Egyptian soil conditions…………..fertilizer used”. These sentences should be more specific than the broad introduction.
Line 15-20: “The experiment……….2020 season”. These lines should be moved from the abstract to the materials and methods section.
Line 21: What exactly do you mean when you say DSM2894? Why is it necessary to use abbreviations in the abstract?
Line 22: Change “compered” to “compared”
Line 23-25: “Phosphorus fertilizer improved protein, anti-radical power, yield as well as nutrient uptake (Fe, Mn, Zn, and Cu) in leaves and roots of sweet potato under CaCO3 stress.” How its improved?
Line 26: What do you mean by 345?
Line 29: which crop performance?
Keywords
“leaf and tuberous root nutrient contents, tuberous root yield” Please consider rewriting these keywords.
Introduction
The introduction does not clearly presents the problem and goals the authors want to achieve with the paper.
Line 33-44: There isn't a single reference in the entire paragraph?? How is this even possible?
Line 46: “According to [1]” Authors should review the journal's reference writing guidelines and make necessary changes.
Line 46-89: These paragraphs are vague; I recommend revising them to be more relevant to your research topic.
Line 102-103: “While this research was motivated ………….semi-arid areas with calcareous soils.” How did you come to this conclusion?
Materials and methods
Line 108: What do you mean by “standard sweet potato”???
Line 114: The reference must be revised in accordance with the journal's guidelines.
Line 126: “Studied treatments….”? It’s unclear.
Section 2.4, 2.5. 2.6 and 2.7 are unclear, suggest rewriting.
Results
Line 177 Change “content” to “contents”
Section 3.2 and 3.3 are unclear, I recommend rewriting your paragraphs and focusing solely on your findings.
Discussion and conclusions
This section is poorly written and should be proofread by someone who is fluent in English, to understand it completely.
References
Please follow the author's guidelines and correct your references as necessary. Journal guidelines are not followed while writing references. For example line 419-421 (reference 4).
Author Response
Agronomy - MDPI
Manuscript ID: agronomy-1641599
Manuscript Title: " Mitigation of CaCO3 Influence on Ipomoea batatas Plants Using Bacillus megaterium DSM 2894 var."
=====================================================================
Dear Ms. Nicu Petrescu
Assistant Editor, MDPI, Romania
        Thank you for your efforts and I’d like also to thank very much the reviewers for their valuable comments. We have corrected the manuscript based on the comments of reviewers, and the corrections made in the text in red color, and are outlined step by step as follows:
Response to the comments of Reviewer 1:
The work "Mitigation of CaCO3 influence on Ipomoea batatas plants using Bacillus megaterium DSM 2894 var." has the potential to be significant, however the text lacks clarity, the methods are poorly defined, the data analysis appears to be inadequate, and the results and discussion presented do not fully support the conclusions. The following are some examples of specific comments that demonstrate these points:
I have found that the authors did not revise this manuscript according to previous comments (as below):
|
NO |
Comment |
Response |
|
1 |
I suggest rewriting the title of the manuscript |
I rewrote the title of manuscript |
|
Abstract |
||
|
2 |
Line 10-14â€Under Egyptian soil conditions … stress on plants†These sentences should be more specific than the broad introduction. |
removed from the abstract in materials and methods (line 38-41) |
|
3 |
Line 15-20: “The experiment … 2020 seasonâ€. These lines should be moved from the abstract to the materials and methods section. |
These lines moved to the materials and methods section (lines 125-127), Two field experiments …….in the 2019 and 2020.  |
|
4 |
Line 21: what exactly do you mean when you say DSM 2894? Why is it necessary to use abbreviations in abstract. |
DSM 2894 is the code number of the isolated strain from Bacillus megaterium. |
|
5 |
Line 22: change “compered†to “compared†|
Changed from “compered†to “compared†line 24 |
|
6 |
Line 23-25: phosphorus fertilizer improved protein, anti-radical power, yield as well as nutrient uptake (Fe, Mn, Zn and Cu) in leaves and roots of sweet potato under CaCO3 stress. How its improved |
The effect of phosphorus fertilizer was mentioned in details as shown (in lines 25-31) |
|
7 |
Line 26: what do you mean by 345? |
345 kg of phosphorus fertilizer applied, known as calcium super phosphate mentioned in line 20. |
|
8 |
Line 29: which crop performance |
deleted “crop performance†from the sentence |
|
Keywords |
||
|
9 |
leaf and tuberous root nutrient contents, tuberous root yield†please consider rewriting these keywords |
I rewrite the keywords again as shown in line 36-37. |
|
Introduction |
||
|
10 |
The introduction dos not clear presents the problem and goals the authors want to achieve with the paper |
The main problems which the research was conducted are mentioned in the first paragraph (lines 38-48) and the last paragraph of the introduction (lines 123-125). |
|
11 |
line 33-44: there isn’t a single reference in the entire |
I added reference inside the first paragraph (line 43). |
|
12 |
Line 46: According to [1]†authors should review the journal’s reference writing guideline and make necessary changes |
The reference changed from [1] to [3] and I cited Wahba et al., 2019 instead of USDA. |
|
13 |
Line 46-89: these paragraphs are vague; I recommend revising them to be more relevant to your research topic |
The introduction section was written again and reviewed by English language editing By MDPI. Â |
|
14 |
Line 102-103: while this research was motivated …… semi-arid areas with calcareous soils.†How did you come to this conclusion. |
I did not mention that this is a conclusion. But I mentioned that there is a possibility that the use of this strain of bacteria may be a reason for increasing the phosphorus availability under calcareous soils, or it may work to reduce the amount of phosphorus fertilizer applied. |
|
Materials and Methods |
||
|
14 |
Line 108: what do you mean by standard sweet potato??? |
I removed it |
|
15 |
Line 114: The reference must be revised in accordance with the journal’s guidelines |
I carefully reviewed, the reference is correct and I changed from according to …. To as described in ….. (line 126) |
|
16 |
Line 126: Studied treatments ….? It’s unclear |
I removed “studied treatments†and adjusted the sentence. |
|
17 |
section 2.4, 2.5, 2.6 and 2.7 are unclear. Suggest rewriting. |
i think the mentioned sections were well illustrated and I modified as shown with the red color. |
|
Results |
||
|
18 |
Line 177 change “content†to “contents†|
Changed from “content†to “contents†in line 183. |
|
19 |
Section 3.2 and 3.3 are unclear. I recommend rewriting your paragraphs and focusing solely on your findings. |
I rewrite sections 3.2 and 3.3 again |
|
Discussion and conclusion |
||
|
20 |
This section is poorly written and should be proofread by someone who is fluent in English to understand it completely |
The discussion and conclusion sections were written again and underwent English language editing by MDPI |
|
References |
||
|
21 |
Please follow the author’s guidelines and correct your references as necessary. Journal guidelines are not followed while writing references. For example line 419-421 (reference 4) |
Done |
Â
Many thanks to Reviewer 1 for his valuable comments
Ahmed A. M. Awad (Corresponding author)
April 3, 2022Â

Reviewer 2 Report
General comments
This manuscript reports on the effects of specific plant growth-promoting rhizobacteria (PGPR) on the field performance of Ipomoea batatas plants in a region of Egypt. Although interesting data were collected, the manuscript has serious deficiencies that render it unsuitable for publication in a high-rank journal like Agronomy. My concerns are summarized below.
Specific comments
Ipomoea batatas in the title (see L2) should be in italics.
There are too many introductory statements in the abstract (see L10-14). This part should be reduced.
The information about the study site, geographical coordinates, experimental design, and replications in the abstract (see L17-18 & L21) is redundant. This part should be reduced.
The abstract should focus on the main findings of the study, instead of background and methodological information.
The introduction does not point out the gap of the literature the study seeks to fill and the novelty of the study over the existing literature. What triggered the study and what is the new information this study brings to the literature?
A relevant hypothesis for the study is missing from the introduction. What was the hypothesis tested in this article? A true scientific question should be formed. This point should be further elaborated.
In Table 1, the names of the variables in the first row are not displayed correctly.
In all Tables, two decimal digits are redundant for all variables. I would suggest rounding at one decimal digit.
In the methodology, more information is required about the establishment of Ipomoea batatas pants to the experimental field.
In the methodology, more information is required about the exact method of application of the PGPR to the plants.
In the methodology, the analysis of the data is not explained clearly. Only the main effects of the treatments are reported. Potential year by treatment interactions are not reported. A table with the analysis of the data is required.
In all Figures, the font size of the labels in x- and y-axis should be increased, so that labels can be visible. The same should be done for the letters of significance on the bars.
The number of figures (totally 22) is quite large for a scientific paper and very tiring for the reader. Some figures could be easily combined to one figure, so that the total number could be reduced. Some others could be deleted without a loss for the paper. An alternative way of presentation should be applied.
All tables are overcrowded. There are too many values in relatively small space. Some values overlap. The reader cannot tell what is what. An alternative way of presentation should be applied.
The novelty of the study should be highlighted in the discussion. Why are the findings significant for the scientific community? What are the trends in other areas of the world? What are the practical implications of the findings in the study area and in other areas?
The authors should incorporate recent papers on the effect of plant growth promoting rhizobacteria on crop growth. For example, please see Journal of Soil Science and Plant Nutrition 19, 592–602 (2019) by Amirnia et al. This paper could be cited in the introduction and in the discussion.
Author Response
Agronomy - MDPI
Manuscript ID: agronomy-1641599
Manuscript Title: " Mitigation of CaCO3 influence on Ipomoea batatas plants using Bacillus megaterium DSM 2894 var."
=====================================================================
Dear Ms. Nicu Petrescu
Assistant Editor, MDPI, Romania
        Thank you for your efforts and I would like also to thank very much the reviewers for their valuable comments. We have corrected the manuscript based on the comments of reviewers, and the corrections made in the text in red color, and are outlined step by step as follows:
Response to the comments of Reviewer 2:
General comments
This manuscript reports on the effects of specific plant growth-promoting rhizobacteria (PGPR) on the field performance of Ipomoea batatas plants in a region of Egypt. Although interesting data were collected, the manuscript has serious deficiencies that render it unsuitable for publication in a high-rank journal like Agronomy. My concerns are summarized below
|
NO |
Comment |
Response |
|
1 |
Ipomoea batatas in the title (see L 2) should be in italics. |
Done |
|
2 |
There are too many introductory statements in the abstract (see L10-14). This part should be reduced. |
Done |
|
3 |
The information about the study site, geographical coordinates, experimental design, and replications in the abstract (see L17-18 & L21) is redundant. This part should be reduced |
Done |
|
4 |
The abstract should focus on the main findings of the study, instead of background and methodological information |
Done |
|
5 |
The introduction does not point out the gap of the literature the study seeks to fill and the novelty of the study over the existing literature. What triggered the study and what is the new information this study brings to the literature? |
I re-wrote the introduction again |
|
6 |
A relevant hypothesis for the study is missing from the introduction. What was the hypothesis tested in this article? A true scientific question should be formed. This point should be further elaborated. |
Done (lines 97-100) |
|
7 |
In Table 1, the names of the variables in the first row are not displayed correctly. |
I mentioned the variable as an abbreviation in the first row and mentioned in detail below the table |
|
8 |
In all Tables, two decimal digits are redundant for all variables. I would suggest rounding at one decimal digit. |
two decimal digits are more accurate in both variable and standard error |
|
9 |
In the methodology, more information is required about the establishment of Ipomoea batatas plants to the experimental field. |
planting method was briefly illustrated (lines 139-140) |
|
10 |
In the methodology, more information is required about the exact method of application of the PGPR to the plants. |
Done (line 134-136) |
|
11 |
In the methodology, the analysis of the data is not explained clearly. Only the main effects of the treatments are reported. Potential year by treatment interactions are not reported. A table with the analysis of the data is required. |
I did not understand this comment. Do you mean that a combined analysis is done and therefore a growth years is considered a study factor? |
|
12 |
In all Figures, the font size of the labels in x- and y-axis should be increased, so that labels can be visible. The same should be done for the letters of significance on the bars |
Done |
|
13 |
The number of figures (totally 22) is quite large for a scientific paper and very tiring for the reader. Some figures could be easily combined to one figure, so that the total number could be reduced. Some others could be deleted without a loss for the paper. An alternative way of presentation should be applied |
22 graphics are not a large number on a scientific paper that took three years to field work on, including cultivation operations in addition to chemical analyzes, especially since the last research paper published in Agronomy (MDPI) last month included 42 graphs and 11 tables
|
|
14 |
All tables are overcrowded. There are too many values in relatively small space. Some values overlap. The reader cannot tell what is what. An alternative way of presentation should be applied |
The tables are not crowded, but the tables can be placed as a landscape instead of portrait |
|
14 |
The novelty of the study should be highlighted in the discussion. Why are the findings significant for the scientific community? What are the trends in other areas of the world? What are the practical implications of the findings in the study area and in other areas? |
mentioned in the conclusion section (lines 639-642) |
|
15 |
The authors should incorporate recent papers on the effect of plant growth promoting rhizobacteria on crop growth. For example, please see Journal of Soil Science and Plant Nutrition 19, 592–602 (2019) by Amirnia et al. This paper could be cited in the introduction and in the discussion |
I think that the number of scientific research related to the manuscript topic is very sufficient and covers the topic very well as described in the revised version
|
The text requires careful revision from a native English-speaking individual. It is not sufficient for an academic publication.
Re: this manuscript was corrected by one of the English editing service listed at https:// www. Mdpi.com/authors/English to check it.
Many thanks to Reviewer 2 for his valuable comments
Ahmed A. M. Awad (Corresponding author)

Reviewer 3 Report
The manuscript " Mitigation of CaCO3 Influence on Ipomoea batatas Plants Us-2 ing Bacillus megaterium DSM 2894 Var.
The methodology applied in the study is appropriate. The results are statistically analyzed and discussed adequately. The present study has important practical implications.However, the manuscript was written carelessly.
line 264 it is: ".....for the non-inoculated sweet potato leaves (DSM0) in both growth seasons,...." it should be: "....for the non-inoculated sweet potato leaves (DSM0) and inoculated sweet potato leaves (DSM1) in both growth seasons,........"
line 271 (title of Tables 4) it is: ".... plants in 2019 and 2020" it should be: ".... plants in 2019 and 2020."
line 276 (under Table 4) it is: ".... phosphorus content and Ca = calcium content . " it should be: "phosphorus content and Ca = calcium content."
line 323 it is: ".... recorded values of 103.58 vs. 110.13 for Mn....." it should be: ".... recorded values of 103.58 vs. 119.51 for Mn....."
line 329 it is: " ... CSP100xBM0 and CSP20xBM0 for Mn and CSP20xBM0 and CSP60xBM0 for Zn..." it should be: " ..... CSP100x DSM0 and CSP20x DSM0 for Mn and CSP20x DSM0 and CSP60x DSM0 for Zn..."
line 406 it is: ".... 13.55 vs. 13.64 for Mn, .... " it should be: ".... 13.55 vs. ????? for Mn,..."
line 437 it is: ".... vs. 4.48 for P in the 2019 and 2020 growth seasons ..." it should be: ".... vs. 4.84 for P in the 2019 and 2020 growth seasons ..."
line 515 it is: " 3.5.2. The effect of phosphorus ..." it should be: " 3.3.2. The effect of phosphorus ..."
line 519 it is: ".... followed by 14.88 vs. 14.771.82 tons ha-1) ..." it should be: "...".... followed by 14.88 vs. 14.77 tons ha-1) ...".
line 634 it is: " .. that DSM 2894 + CPS can be used.." it shoul be: .." that DSM 2894 + CSP can be used
in Table 3 it is: "345 kg ha-1 of calcium super phosphate (100% of RPF) with incubated plants by Bacillus megaterium DSM 2894 var. " it should be: "345 kg ha-1 of calcium super phosphate (100% of RPF) non-inoculated plants by Bacillus megaterium DSM 2894 var ".
in the row of Table 5 it is. " Tubers (mg kg-1) " it should be: "Leaves (mg kg-1) ".
there is no row in table 8 with DSM0, DSM1
error in Figs. and in title of Figs.: 5,6,7,8,13,14,15,16,19,20 it is "DMS1" it should be: " DSM1"
error in Fig 21. it is: " Pf or FP.." it shloul be: " CSP20 CSP40 ....
error in Fig. 22 it is "BMS0 , BMS1" it should be: " DSM0, DSM1 "
Author Response
Agronomy - MDPI
Manuscript ID: Agronomy-1641599
Manuscript Title: "Mitigation of CaCO3 Influence on Ipomoea batatas Plants Using Bacillus megaterium DSM 2894 var."
=====================================================================
Dear Ms. Nicu Petrescu
Assistant Editor, MDPI, Romania
        Thank you for your efforts and I’d like also to thank very much the reviewers for their valuable comments. We have corrected the manuscript based on the comments of reviewers, and the corrections made in the text in red color, and are outlined step by step as follows:
Response to the comments of Reviewer 3:
|
NO |
Comment |
Response |
|
1 |
The methodology applied in the study is appropriate. The results are statistically analyzed and discussed adequately. The present study has important practical implications. However, the manuscript was written carelessly. |
Thank you so much for your opinion and I’d like to inform you that the manuscript was rewrote again and underwent English language editing by MDPI |
|
1 |
Line 264 it is: ".....for the non-inoculated sweet potato leaves (DSM0) in both growth seasons,...." it should be: "....for the non-inoculated sweet potato leaves (DSM0) and inoculated sweet potato leaves (DSM1) in both growth seasons,........" |
Done |
|
2 |
Line 271 (title of Tables 4) it is: ".... plants in 2019 and 2020" it should be: ".... plants in 2019 and 2020." |
Done |
|
3 |
Line 276 (under Table 4) it is: ".... phosphorus content and Ca = calcium content . " it should be: "phosphorus content and Ca = calcium content." |
Done |
|
4 |
line 323 it is: ".... recorded values of 103.58 vs. 110.13 for Mn....." it should be: ".... recorded values of 103.58 vs. 119.51 for Mn....." |
Done (line 325) |
|
5 |
line 329 it is: " ... CSP100xBM0 and CSP20xBM0 for Mn and CSP20xBM0 and CSP60xBM0 for Zn..." it should be: " ..... CSP100x DSM0 and CSP20x DSM0 for Mn and CSP20x DSM0 and CSP60x DSM0 for Zn..." |
Done |
|
6 |
line 406 it is: ".... 13.55 vs. 13.64 for Mn, .... " it should be: ".... 13.55 vs. ????? for Mn,..." |
After reviewing the values of the statistical analysis during the two seasons, the values are is written 13.55 vs. 13.64 for Mn |
|
7 |
Line 437 it is: ".... vs. 4.48 for P in the 2019 and 2020 growth seasons ..." it should be: ".... vs. 4.84 for P in the 2019 and 2020 growth seasons ..." |
Done in line 440 |
|
8 |
line 515 it is: " 3.5.2. The effect of phosphorus ..." it should be: " 3.3.2. The effect of phosphorus ..." |
Done |
|
9 |
line 519 it is: ".... followed by 14.88 vs. 14.771.82 tons ha-1) ..." it should be: "...".... followed by 14.88 vs. 14.77 tons ha-1) ..." |
Done in line 522 |
|
10 |
line 634 it is: " .. that DSM 2894 + CPS can be used.." it should be: .." that DSM 2894 + CSP can be used |
Done in line 639 |
|
11 |
in Table 3 it is: "345 kg ha-1 of calcium super phosphate (100% of RPF) with incubated plants by Bacillus megaterium DSM 2894 var. " it should be: "345 kg ha-1 of calcium super phosphate (100% of RPF) non-inoculated plants by Bacillus megaterium DSM 2894 var ". |
Done |
|
12 |
In the row of Table 5 it is. " Tubers (mg kg-1) " it should be: "Leaves (mg kg-1) " |
Done |
|
13 |
there is no row in table 8 with DSM0, DSM1 |
Done |
|
14 |
error in Figs. and in title of Figs.: 5,6,7,8,13,14,15,16,19,20 it is "DMS1" it should be: " DSM1" |
Done |
|
14 |
error in Fig 21. it is: " Pf or FP.." it should be: " CSP20 CSP40 .... |
Done |
|
15 |
error in Fig. 22 it is "BMS0 , BMS1" it should be: " DSM0, DSM1 " |
Done |
Â
Many thanks to Reviewer 3 for his valuable comments
Ahmed A. M. Awad (Corresponding author)
April 3, 2022

Round 2
Reviewer 2 Report
Authors did not respond to my major remarks and did not follow advice how to improve the manuscript. According to the revised version of the manuscript it seems that they disagree with my opinion but I did not obtain any explanation.